# RoME: Domain-Robust Mixture-of-Experts for MILP Solution Prediction across Domains

**Tianle Pu[1], Zijie Geng[2], Haoyang Liu[2], Shixuan Liu[3,1], Jie Wang[2], Li Zeng[1],**
**Chao Chen[1]\*, Changjun Fan[1]\***

[1]Laboratory for Big Data and Decision, College of Systems Engineering,
National University of Defense Technology
[2]MoE Key Laboratory of Brain-inspired Intelligent Perception and Cognition,
University of Science and Technology of China
[3]College of Computer Science and Technology, National University of Defense Technology
{putl22, liushixuan, zengli24, chenc1997, fanchangjun}@nudt.edu.cn
{zijiegeng, dgyoung}@mail.ustc.edu.cn, jiewangx@ustc.edu.cn

## Abstract

Mixed-Integer Linear Programming (MILP) is a fundamental and powerful framework for modeling complex optimization problems across diverse domains. Recently, learning-based methods have shown great promise in accelerating MILP solvers by predicting high-quality solutions. However, most existing approaches are developed and evaluated in single-domain settings, limiting their ability to generalize to unseen problem distributions. This limitation poses a major obstacle to building scalable and general-purpose learning-based solvers. To address this challenge, we introduce **RoME**, a domain-**Ro**bust **M**ixture-of-**E**xperts framework for predicting MILP solutions across domains. RoME dynamically routes problem instances to specialized experts based on learned task embeddings. The model is trained using a two-level distributionally robust optimization strategy: inter-domain to mitigate global shifts across domains, and intra-domain to enhance local robustness by introducing perturbations on task embeddings. We reveal that cross-domain training not only enhances the model's generalization capability to unseen domains but also improves performance within each individual domain by encouraging the model to capture more general intrinsic combinatorial patterns. Specifically, a single RoME model trained on three domains achieves an average improvement of $67.7\%$ then evaluated on five diverse domains. We further test the pretrained model on MIPLIB in a zero-shot setting, demonstrating its ability to deliver measurable performance gains on challenging real-world instances where existing learning-based approaches often struggle to generalize.

## 1 Introduction

Mixed-Integer Linear Programming (MILP) is a fundamental and expressive framework for modeling a wide range of real-world optimization problems, including logistics optimization [1], network design [2], scheduling [3], and industrial planning [4]. Due to its strong modeling capacity, MILP has become a cornerstone of operation research and combinatorial optimization [5–7]. However, MILPs are well known to be NP-hard, and solving large-scale instances with thousands of variables and constraints remains computationally intensive. Even state-of-the-art solvers such as Gurobi [8] and SCIP [9], which integrate advanced techniques like branch-and-bound, cutting planes, and heuristics, can struggle to deliver timely and scalable solutions as problem complexity grows.

---

*Corresponding author.

39th Conference on Neural Information Processing Systems (NeurIPS 2025).

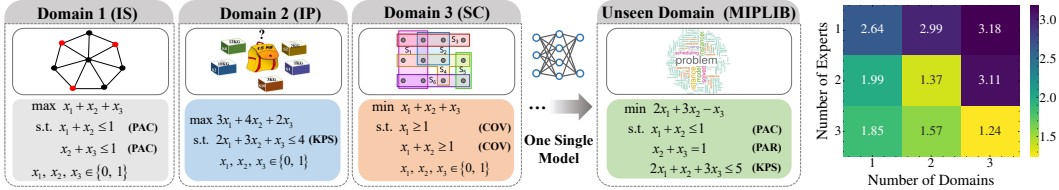

(a) Cross-Domain Training and Generalization          (b) Objective gap (%)

Figure 1: **Illustration of cross-domain training.** (a) A single model is trained on a collection of MILP domains with varying constraint types [10], such as PAC (set packing), KPS (knapsack), COV (set covering), and PAR (set partitioning), and evaluated on unseen domains such as MIPLIB. (b) We conduct experiments across five distinct domains, reporting the average objective gap in percentage relative to the best-known solution while varying the number of experts and training domains. Each value in the heatmap denotes the average percentage gap to the best-known solution. The results show that training on more diverse domains and selecting an appropriate number of experts can effectively reduce this objective gap.

To alleviate these limitations, recent research has explored the use of machine learning to accelerate MILP solving by identifying data-driven patterns in problem instances [11–14]. These approaches fall broadly into two categories. The first integrates neural networks into solver internals by replacing key components such as variable selection [11], cutting plane selection [12], and neighborhood destruction strategies in large neighborhood search [15, 16]. The second line of research, which we focus on in this paper, aims to directly predict initial feasible solutions [17–19], which are then refined by solvers to improve solution quality. These methods have shown promising results on large-scale MILPs and are gaining increasing attention. However, while this solution-prediction paradigm has shown strong empirical performance, most existing methods are trained and evaluated within single-domain settings, which presents two major challenges. First, the learned models often overfit to domain-specific patterns rather than capturing generalizable combinatorial structures, leading to limited generalization to unseen domains. Second, deploying such models in practice often requires collecting new training data for each specific scenario, limiting scalability and broader applications.

To build scalable learning-based solvers that generalize across domains, we advocate for a paradigm shift from single-domain training to frameworks that explicitly embrace cross-domain structural diversity. An illustration of the cross-domain training task is in Figure 1. A central insight of this work is that training on multiple MILP domains not only improves generalization under distribution shift but also enhances solution quality within each individual domain (see Section 4.2). Rather than treating distributional differences as noise, we view them as valuable signals that help the model capture transferable combinatorial patterns. In this light, cross-domain training naturally acts as a form of domain-level regularization, steering the model away from memorization and towards learning general principles that transfer across tasks.

In this paper, we propose RoME, a Domain-**Ro**bust **M**ixture-of-**E**xperts framework for MILP solution prediction. RoME aims to generalize effectively across various problem distributions while maintaining strong performance on specific tasks. The framework integrates a Mixture-of-Experts (MoE) architecture with a two-level Distributionally Robust Optimization (DRO) objective into a unified trainable framework. The MoE consists of multiple expert networks, each specializing in distinct MILP distributions. For each instance, the MoE dynamically routes it to specialized experts based on its graph embedding, allowing the model to adapt both its internal representations and final outputs to the unique structural characteristics of each instance. To further enhance robustness, RoME is trained using a two-level distributionally robust optimization strategy. (1) Inter-domain robustness is achieved through group-DRO, which minimizes the worst-case loss across domains, thereby reducing the impact of global distributional shifts. (2) Intra-domain robustness is ensured by applying isotropic perturbations to the task embeddings, which stabilizes expert selection and predictions.

To demonstrate the effectiveness of RoME, we conduct comprehensive experiments to demonstrate the effectiveness of RoME. Specifically, we train a single RoME model on three domains and evaluate it across five diverse domains, where it achieves an average improvement of $67.7\%$ over strong baselines. We further test the pretrained model on MIPLIB in a zero-shot setting, demonstrating a significant performance gain on challenging instances that existing learning-based approaches often struggle to generalize. Finally, to gain a deeper understanding of RoME, we conduct interpretability

analyses that find expert activation patterns and task embeddings, shedding light on how the model captures and transfers structural knowledge across domains.

## 2 Related Work

### 2.1 Machine Learning for MILPs

Machine learning has been widely applied to the field of MILPs [20–23]. Recent work on learning-based MILP solvers can be roughly classified into two categories. The first line of research integrates machine learning to enhance the solving efficiency of conventional solvers. Researchers utilize a learned model to replace some key modules in the solvers, including variable selection [11, 24, 13], node selection [25–27], cutting plane selection [12, 28] and large neighborhood search [15, 16, 29–31]. Another line of research focuses on leveraging learning-based models to predict an initial solution for MILPs. Nair et al. [17] introduces the first solution-prediction framework for MILPs. Building on this foundation, Han et al. [18] proposes the Predict-and-Search (PS) framework, which incorporates a trust-region search to improve feasibility and solution quality. To further enhance prediction accuracy, Huang et al. [19] employs contrastive learning to train the solution-prediction network. Liu et al. [32] adopts a prediction–correction paradigm to obtain higher-quality solutions by iteratively refining initially mis-predicted variable assignments. Notably, Geng et al. [33] is the first to propose an unsupervised method that incorporates gradient information into the prediction model for MILPs, thereby significantly reducing the cost of collecting high-quality training data.

### 2.2 Cross-Domain Training Techniques

Mixture of Experts (MoE) [34] and Distributionally Robust Optimization (DRO) [35, 36] are two core methodologies for cross-domain training, where MoE gives the model with the ability to perceive and adapt to different domains, and DRO enhances robustness by balancing training across domains. Early studies [37] focus on designing dedicated experts for individual domains, while recent research [38, 39] has shifted toward sparse architectural variants. The MoE paradigm has been successfully applied to graph-based combinatorial optimization problems. For instance, MVMoE [40] tackles a suite of vehicle-routing variants, and MoE-style encoder-decoder frameworks, e.g., MAB-MTL [41], GCNCO [42], GOAL [43], consistently adopt a "header-encoder-decoder" architecture. In contrast, DRO guarantees robustness to potential domain shifts by minimizing the worst-case risk over an uncertainty set. Sagawa et al. [44] first introduces Group DRO, which balances the losses across different groups. CCD-DG [45] builds a class-conditioned Wasserstein ball and automatically tunes its radius to protect against conditional shifts, while Moderately-DRO [46] and its stochastic variant [47] improves exploration of unseen domains and offeres tighter generalization bounds with lower sampling complexity.

## 3 Methodology

In this section, we present **RoME**, a domain-**Ro**bust **M**ixture-of-**E**xperts framework for cross-domain MILP prediction. This section is organized as follows. Section 3.1 introduces the problem formulation, the predict-and-search (PS) framework, and the cross-domain learning settings. Section 3.2 presents the Mixture-of-Experts (MoE) architecture for instance-adaptive structure-awareness across different domains. Section 3.3 presents a robust training objective that effectively improves the robustness of the MoE model training. Finally, Section 3.4 introduces a group-level Distributionally Robust Optimization (DRO) scheme to handle the cross-domain training process.

### 3.1 Problem Formulation

We focus on the task of learning to predict high-quality solutions for Mixed-Integer Linear Programming (MILP) problems. A MILP instance $\mathcal{I}$ is defined as:

$$\min_{\boldsymbol{x} \in \mathbb{Z}^p \times \mathbb{R}^{n-p}} \left\{ \boldsymbol{c}^\top \boldsymbol{x} \mid \boldsymbol{A}\boldsymbol{x} \leq \mathbf{b},\, \boldsymbol{l} \leq \boldsymbol{x} \leq \boldsymbol{u} \right\}, \tag{1}$$

where $\boldsymbol{x} \in \mathbb{R}^n$ denotes the decision variables, with the first $p$ entries being integer and the remaining $n - p$ continuous. The vector $\boldsymbol{c} \in \mathbb{R}^n$ denotes the coefficients of the linear objective, the constraints

are defined by the matrix $\boldsymbol{A} \in \mathbb{R}^{m \times n}$ and the righ-hand side vector $\boldsymbol{b} \in \mathbb{R}^m$, and the variable bounds are given by $\boldsymbol{l} \in (\mathbb{R} \cup \{-\infty\})^n$ and $\boldsymbol{u} = (\mathbb{R} \cup \{+\infty\})^n$. Without loss of generality, we focus on binary integer variables, and general integer variables can be handled via standard preprocessing techniques [17].

We encode each MILP instance as a bipartite graph $\mathcal{G} = (\mathcal{W} \cup \mathcal{V}, \mathcal{E})$, where $\mathcal{W}$ and $\mathcal{V}$ denote the sets of constraint and variable nodes, respectively, and the edge set $\mathcal{E}$ corresponds to non-zero entries in $\boldsymbol{A}$. Each node and edge is associated with a set of features derived from problem coefficients and structural attributes. Such a bipartite graph can completely describe a MILP instance, enabling graph neural networks (GNNs) to process the instances and predict their solutions.

We adopt the Predict-and-Search (PS) paradigm to approximate the solution distribution of a given MILP. Specifically, the distribution is defined via an energy function that assigns lower energy to high-quality feasible solutions and infinite energy to infeasible ones:

$$p(\boldsymbol{x} \mid \mathcal{I}) = \frac{\exp(-E(\boldsymbol{x} \mid \mathcal{I}))}{\sum_{\boldsymbol{x}'} \exp(-E(\boldsymbol{x}' \mid \mathcal{I}))}, \quad \text{where} \quad E(\boldsymbol{x} \mid \mathcal{I}) = \begin{cases} \boldsymbol{c}^\top \boldsymbol{x}, & \text{if } \boldsymbol{x} \text{ is feasible,} \\ +\infty, & \text{otherwise.} \end{cases} \quad (2)$$

Our goal is to learning distribution $p_{\boldsymbol{\theta}}(\boldsymbol{x}|\mathcal{I})$ for a given instance $\mathcal{I}$. To make learning tractable, we assume a fully factorized solution distribution over the binary variables, i.e., $p_{\boldsymbol{\theta}}(\boldsymbol{x}|\mathcal{I}) = \prod_{i=1}^{p} p_{\boldsymbol{\theta}}(x_i|\mathcal{I})$, where $p_{\boldsymbol{\theta}}(x_i|\mathcal{I})$ denotes the predicted marginal probability for variable $x_i$. To do so, we use a GNN model to output a $p$-dimension vector $\hat{\boldsymbol{x}} = \boldsymbol{f}_{\boldsymbol{\theta}}(\mathcal{I}) = (\hat{x}_1, \cdots, \hat{x}_p)^\top \in [0,1]^p$, where $\hat{x}_j = p_{\boldsymbol{\theta}}(x_j = 1|\mathcal{I})$. To train the model, we use a weighted set of feasible solutions $\{\boldsymbol{x}^{(i)}\}_{i=1}^{N}$ as supervised signals, where each solution is assigned a weight $w_i \propto \exp(-\boldsymbol{c}^\top \boldsymbol{x}^{(i)})$. We let $x_j^{(i)} = p_{\boldsymbol{\theta}}(x_j^{(i)} = 1|\mathcal{I})$. Then, the training loss function is a binary cross-entropy loss defined as:

$$\mathcal{L}_{\text{BCE}}(\boldsymbol{\theta}|\mathcal{I}) = -\sum_{i=1}^{N} \sum_{j=1}^{p} w_i \cdot \left[ x_j^{(i)} \log \hat{x}_j + (1 - x_j^{(i)}) \log(1 - \hat{x}_j) \right]. \quad (3)$$

At inference time, the GNN model outputs a predicted marginals $\hat{\boldsymbol{x}} \in [0,1]^p$. A standard MILP solver, e.g., Gurobi or SCIP is then used to search for a feasible solution in a local neighborhood around $\hat{\boldsymbol{x}}$ by solving the following trust region problem:

$$\min_{\boldsymbol{x} \in \mathbb{Z}^p \times \mathbb{R}^{n-p}} \left\{ \boldsymbol{c}^\top \boldsymbol{x} \mid \boldsymbol{A}\boldsymbol{x} \leq \mathbf{b}, \; \boldsymbol{l} \leq \boldsymbol{x} \leq \boldsymbol{u}, \boldsymbol{x}_{1:p} \in \mathcal{B}(\hat{\boldsymbol{x}}, \Delta) \right\}, \quad (4)$$

where the trust region $\mathcal{B}(\hat{\boldsymbol{x}}, \Delta) := \{\boldsymbol{x} \in \mathbb{R}^n : \|\boldsymbol{x}_{1:p} - \hat{\boldsymbol{x}}\|_1 \leq \Delta\}$ constrains the solver to remain close to the predicted binary configuration..

It's evident that the quality of the final solution depends heavily on the accuracy of the predicted marginals. Poor predictions can mislead the solver and limit optimization performance. Therefore, we focus on training a predictive model $\boldsymbol{f}_{\boldsymbol{\theta}}$ that can accurately predict the marginal probabilities.

### 3.2 Mixture-of-Experts with Structure-Aware Routing

We consider the setting where training samples are drawn from $K$ different domains $\{\mathcal{D}_1, \ldots, \mathcal{D}_K\}$, each corresponding to a structurally distinct MILP family. Our goal is to learn a single unified model that can robustly generalize not only across these diverse domains, but also those domains unseen during training. However, due to the significant combinatorial differences between domains, capturing all structural variations within a single monolithic model is challenging.

To address this challenge, we adopt a Mixture-of-Experts (MoE) framework that comprises three key components: a shared graph encoder, multiple expert networks, and a task decoder. The shared graph encoder learns common structural representations across tasks, while the expert networks capture task-specific decision patterns under the guidance of a dynamic gating mechanism that adaptively assigns expert weights based on task-level embeddings. Finally, the task decoder integrates the aggregated expert outputs to generate the final prediction. This design enables structure-aware representation learning and output specialization.

**Shared Graph Encoder** Given a MILP instance $\mathcal{I}$, represented as a bipartite graph $\mathcal{G}$, we apply a graph neural network (GNN) to extract node embeddings $\{\boldsymbol{h}_1, \ldots, \boldsymbol{h}_p\} \subset \mathbb{R}^d$, where $d$ is the dimension of the hidden space. We then compute a global graph embedding $\boldsymbol{h}_{\mathcal{G}} \in \mathbb{R}^d$ by mean pooling over variable nodes, i.e., $\boldsymbol{h}_{\mathcal{G}} = \frac{1}{p} \sum_{j=1}^{p} \boldsymbol{h}_j$. In this work, we follow the GNN encoder design used in [18].

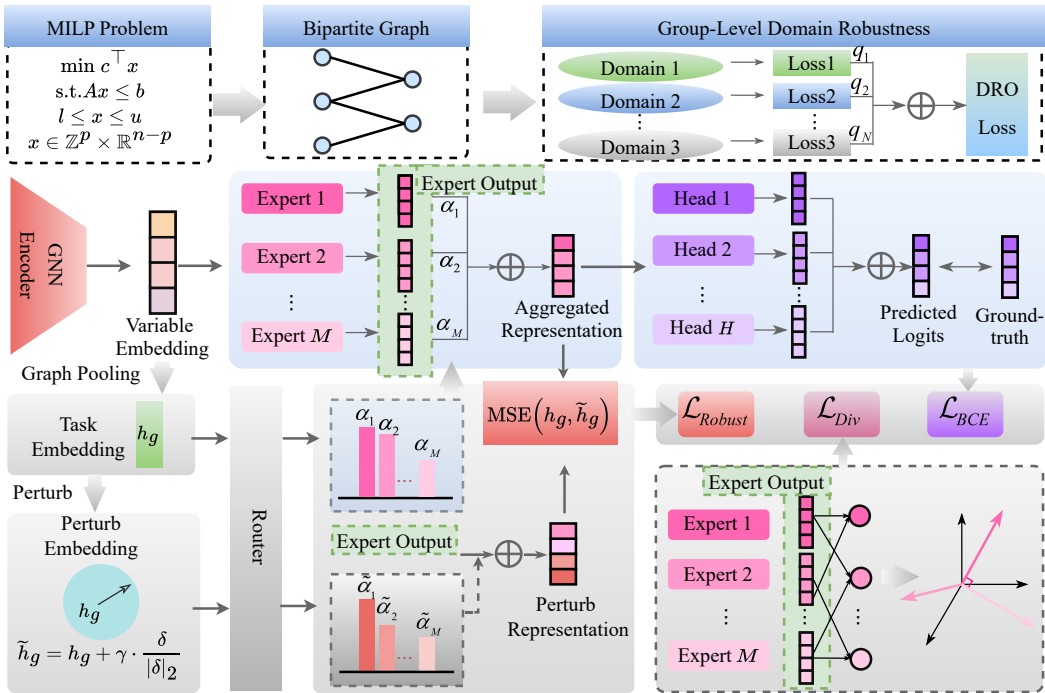

Figure 2: **The overview of RoME.** RoME employ a MoE architecture with structure-aware representation learning. For cross-domain training, RoME proposes a robust training object containing binary cross-entropy loss $\mathcal{L}_{\text{BCE}}$, expert diversity regularization $\mathcal{L}_{\text{Div}}$ and fuzzy membership consistency loss $\mathcal{L}_{\text{Robust}}$. To further enhance generalization across structurally diverse MILP families, RoME uses a group-level domain robust scheme to construct DRO loss.

**Multiple Expert Network** To capture structural diversity, we define a set of $M$ expert networks $\{\boldsymbol{f}_{\boldsymbol{\theta}}^{(1)}, \cdots, \boldsymbol{f}_{\boldsymbol{\theta}}^{(M)}\}$, where each expert $\boldsymbol{f}_{\boldsymbol{\theta}}^{(m)} : \mathbb{R}^d \to \mathbb{R}^d$ maps the variable embeddings into an expert-specific representation space:

$$\boldsymbol{z}_j^{(m)} = \boldsymbol{f}_{\boldsymbol{\theta}}^{(m)}(\boldsymbol{h}_j). \tag{5}$$

Each expert can be interpreted as modeling a different mode of structural regularity, allowing the model to specialize across domains. To route instances to experts, we use a gating network $\boldsymbol{g}_{\boldsymbol{\theta}}^{\text{enc}} : \mathbb{R}^d \to \mathbb{R}^M$ that maps the graph-level embedding $\boldsymbol{h}_{\mathcal{G}}$ into a soft weighting over all experts:

$$\boldsymbol{\alpha} = \text{Softmax}\left(\boldsymbol{g}_{\boldsymbol{\theta}}^{\text{enc}}(\boldsymbol{h}_{\mathcal{G}})/\tau\right), \tag{6}$$

where $\tau$ is a temperature parameter controlling the sharpness of the routing distribution. The vector $\boldsymbol{\alpha} \in [0,1]^M$ can be interpreted as softly indicating which experts are most relevant for the current instance. The final representation for each variable is computed by aggregating the expert outputs:

$$\boldsymbol{z}_j = \sum_{m=1}^{M} \alpha_m \cdot \boldsymbol{f}_{\boldsymbol{\theta}}^{(m)}(\boldsymbol{h}_j) \tag{7}$$

**Task Decoder** To further enhance specialization, we adopt a multi-head decoder that enables the model to learn and combine multiple task-specific representations. Specifically, we use the decoder set $\{d_{\boldsymbol{\theta}}^{(1)}, \cdots, d_{\boldsymbol{\theta}}^{(H)}\}$ with $H$ decoders, each mapping $\boldsymbol{z}_j$ to a scalar logit:

$$\sigma_j^{(h)} = d_{\boldsymbol{\theta}}^{(h)}(\boldsymbol{z}_j). \tag{8}$$

Another gating network $\boldsymbol{g}_{\boldsymbol{\theta}}^{\text{dec}}$—which shares a similar architecture but different parameters with the encoder gate—produces a soft weighting vector $\boldsymbol{\beta} \in [0,1]^H$ over the decoders:

$$\boldsymbol{\beta} = \text{Softmax}\left(g_{\boldsymbol{\theta}}^{\text{dec}}(\boldsymbol{h}_{\mathcal{G}})/\tau'\right), \tag{9}$$

where $\tau'$ is a temperature parameter controlling the sharpness of the routing distribution. The final logit is obtained by aggregating the outputs from all decoder heads:

$$\sigma_j = \sum_{h=1}^{H} \beta_h \cdot \sigma_j^{(h)}, \tag{10}$$

and the predicted marginal is given by

$$\hat{x}_j = \mathrm{Sigmoid}(\sigma_j). \tag{11}$$

This flexible MoE design allows the model to dynamically adjust both internal representations and final outputs to match the structural characteristics of each instance. The expert network captures variable-level diversity, while the task decoder adds output-level specialization to further enhance generalization across domains.

## 3.3 Intra-Domain Robust Training Objective

To enable intra-domain robustness across structurally diverse instances in a domain, we introduce a composite loss function consisting of three components: a binary cross-entropy loss to supervise the training and a regularization term to promote output stability under domain uncertainty. Formally, for a given training instance $\mathcal{I}$, the total objective is given by:

$$\mathcal{L}(\boldsymbol{\theta}|\mathcal{I}) = \mathcal{L}_{\mathrm{BCE}} + \lambda_{\mathrm{Div}} \cdot \mathcal{L}_{\mathrm{Div}} + \lambda_{\mathrm{Robust}} \cdot \mathcal{L}_{\mathrm{Robust}}, \tag{12}$$

where $\lambda_{\mathrm{Div}}$ and $\lambda_{\mathrm{Robust}}$ are hyperparameters balancing these terms.

**Binary Cross-Entropy Loss $\mathcal{L}_{\mathbf{BCE}}$.** As introduced in Section 3.1, we train the model by minimizing the weighted binary cross-entropy (3) between the predicted marginals $p_\theta(x_j = 1 \mid \mathcal{I})$ and a weighted set of feasible solutions sampled from the MILP instance. This loss serves as the primary signal for learning meaningful solution distributions.

**Expert Diversity Regularization $\mathcal{L}_{\mathbf{Div}}$.** To encourage each expert to learn distinct behaviors and avoid mode collapse, we regularize the similarity among expert outputs. Recall that $\boldsymbol{z}_j^{(m)}$ denote the output embedding of expert $m$ for variable $j$ after the expert network layer. Let $\boldsymbol{z}^{(m)}$ denote the concatenation of the representation of all variables, i.e., $\boldsymbol{z}^{(m)} = \mathrm{Concat}(\boldsymbol{z}_1^{(m)}, \cdots, \boldsymbol{z}_p^{(m)})$. The diversity loss is then computed as:

$$\mathcal{L}_{\mathrm{Div}} = \frac{1}{M(M-1)} \sum_{m \neq m'} \frac{\left| \left\langle \boldsymbol{z}^{(m)}, \boldsymbol{z}^{(m')} \right\rangle \right|}{\left\| \boldsymbol{z}^{(m)} \right\| \cdot \left\| \boldsymbol{z}^{(m')} \right\|}. \tag{13}$$

This term encourages orthogonal expert representations, promoting more diverse perspectives across MILP structures.

**Fuzzy Membership Consistency Loss $\mathcal{L}_{\mathbf{Robust}}$.** To improve the model's robustness to domain uncertainty, we adopt a fuzzy membership view of the expert routing mechanism. The routing vector $\boldsymbol{\alpha} = \boldsymbol{g}_\theta(\boldsymbol{h}_\mathcal{G}) \in [0,1]^M$, derived from the task embedding $\boldsymbol{h}_\mathcal{G}$, can be interpreted as a soft domain membership, indicating the degree to which each expert should be activated. However, for unseen or ambiguous MILP instances, the true domain identity may be uncertain or poorly represented in the training data. In such cases, small variations in the task embedding may lead to unstable routing decisions and inconsistent predictions. To address this, we encourage the model to maintain output consistency under slight changes in the domain assignment. Concretely, we apply an isotropic perturbation to the task embedding:

$$\tilde{\boldsymbol{h}}_\mathcal{G} = \boldsymbol{h}_\mathcal{G} + r \cdot \frac{\boldsymbol{\delta}}{\|\boldsymbol{\delta}\|_2}, \quad \boldsymbol{\delta} \sim \mathcal{N}(\boldsymbol{0}, \boldsymbol{I}), \tag{14}$$

where $r$ is a learnable scalar controlling the perturbation magnitude. This perturbed task embedding induces a new routing vector $\tilde{\boldsymbol{\alpha}} = \boldsymbol{g}_\theta(\tilde{\boldsymbol{h}}_\mathcal{G})$, which results in a different mixture of expert representations $\tilde{\boldsymbol{z}}_j$. We then enforce consistency between the original and perturbed representations using the following loss:

$$\mathcal{L}_{\mathrm{Robust}} = \frac{1}{p} \sum_{j=1}^{p} \|\tilde{\boldsymbol{z}}_j - \boldsymbol{z}_j\|_2^2. \tag{15}$$

This regularization encourages the model to produce stable representations even under uncertainty in domain assignment, thus improving its robustness to both domain shift and embedding noise. Beyond its regularization role, this loss is also theoretically motivated from two complementary perspectives. First, from a computational-complexity viewpoint, most MILP classes are NP-complete, implying that different MILPs can, in principle, be reduced to one another, a fact confirmed by recent advances in graph-based combinatorial optimization [5, 48]. Building on this insight, we exploit structural similarities among domains so that related problems route to overlapping expert subsets, enhancing cross-domain generalization. Perturbing task embeddings partially simulates such cross-domain shifts, enforcing the same expert outputs for perturbed and original embeddings strengthens this alignment and encourages domain-invariant reasoning. Second, from an adversarial-training viewpoint, embedding perturbations stabilize the training, helping RoME capture high-level patterns that persist across domains.

### 3.4 Inter-Domain Group-Level Robustness

To further enhance generalization across structurally diverse MILP families, we introduce a domain-level training strategy based on distributionally robust optimization (DRO). Rather than minimizing a uniform average of per-domain losses, we adopt a min-max objective that optimizes performance under the worst-case distribution over training domains. Formally, let $\mathcal{L}_k$ denote the expected loss on domain $\mathcal{D}_k$, estimated empirically via mini-batch sampling:

$$\mathcal{L}_k(\boldsymbol{\theta}) := \mathbb{E}_{\mathcal{I} \sim \mathcal{D}_k}\left[\mathcal{L}(\boldsymbol{\theta}|\mathcal{I})\right], \tag{16}$$

where $\mathcal{L}(\boldsymbol{\theta}|\mathcal{I})$ is the instance-level training loss as defined in Section 3.3. We introduce a probability vector $\boldsymbol{q} \in \Delta_K$, where $\Delta_K$ denotes the probability simplex over the $K$ domains, satisfying $\sum_{k=1}^{K} q_k = 1$ and $q_k \geq 0$. We aim to optimize the following objective:

$$\min_{\theta} \sup_{\boldsymbol{q} \in \Delta_K} \sum_{k=1}^{K} q_k \cdot \mathcal{L}_k(\boldsymbol{\theta}), \tag{17}$$

which seeks to minimize the worst-case expected loss under all convex combinations of domains-specific losses. To achieve this, we alternatively update $\boldsymbol{\theta}$ and $\boldsymbol{q}$. Intuitively, we maintain a distribution $\boldsymbol{q}$ over groups, with high masses on high-loss groups, and update on each example proportionally to the mass on its group. In each iteration, we first estimate $\mathcal{L}_k$ for a randomly chosen domain using a minibatch. Then, we update the domain weights $\boldsymbol{q}$ using exponential gradient ascent so that $q_k \propto \exp(\eta \cdot \mathcal{L}_k)$, where $\eta > 0$ is a hyperparameter controlling the sensitivity to high-loss domains. The resulting weights $\boldsymbol{q}$ are then normalized to form a valid probability vector over domains. Therefore, domains with higher empirical losses are thus assigned larger weights, allowing the model to focus learning capacity on harder or underrepresented domains. By coupling this domain-level reweighting with the instance-level regularization objectives introduced in Section 3.3, our training pipeline enables scalable and principled optimization for robust multi-domain MILP learning.

## 4 Experiments

We conduct comprehensive experiments to evaluate the effectiveness of RoME in learning a unified model for solving MILP problems from several different domains, including those in-distribution and out-of-distribution datasets, and zero-shot generalization to real-world instances from MIPLIB. We then conduct experiments to analyze the contributions of different components, as well as the featured patterns of our learned model. All experiments are conducted on a single machine with NVIDIA GeForce RTX 3090 GPUs and 24-core AMD EPYC 7402 CPUs (2.80 GHz). Code is available at `https://github.com/happypu326/RoME`. More implementation details can be found in Appendix A.

### 4.1 Experimental Setup

**Datasets** We conduct experiments on five MILP problem families widely studied in the literature: Independent Set (IS), Set Covering (SC), Item Placement (IP), Combinatorial Auctions (CA), and Workload Appointment (WA). The instances in the dataset have diverse kinds of combinatorial structures and constraints, with different levels of constraint coefficient sparsity, variable numbers, and objective complexity. For IS SC and CA, we follow the conventional instances generation process in existing works [11, 12, 18, 19]. The IP and WA datasets are from NeurIPS 2021 ML4CO competition [49]. Please see Appendix C for more detailed information on the benchmarks.

Table 1: Performance comparison across various MILP domains, under a $1,000$s time limit. We train RoME on IS, IP and SC, while the performance of RoME on WA and CA is zero-shot performance. '↑' indicates that higher is better, and '↓' indicates that lower is better. We mark the **best values** in bold. We also report the improvement of our method over the traditional solvers in terms of $\text{gap}_{\text{abs}}$.

| | IS (BKS 685.00) | | IP (BKS 11.16) | | SC (BKS 124.64) | | WA (BKS 703.05) | | CA (BKS 97524.37) | |
|---|---|---|---|---|---|---|---|---|---|---|
| | Obj ↑ | Time ↓ | Obj ↓ | $\text{gap}_{\text{abs}}$ ↓ | Obj ↓ | $\text{gap}_{\text{abs}}$ ↓ | Obj ↓ | $\text{gap}_{\text{abs}}$ ↓ | Obj ↑ | $\text{gap}_{\text{abs}}$ ↓ |
| Gurobi | 685.00 | 57.35 | 11.43 | 0.27 | 125.21 | 0.57 | 703.47 | 0.42 | 97308.93 | 215.44 |
| PS+Gurobi | 685.00 | 14.67 | 11.40 | 0.24 | 125.17 | 0.53 | 703.47 | 0.42 | 97280.70 | 243.67 |
| ConPS+Gurobi | 685.00 | 20.13 | 11.36 | 0.20 | 125.18 | 0.54 | 703.47 | 0.42 | 97395.83 | 128.54 |
| RoME+Gurobi | **685.00** | **2.08** | **11.22** | **0.06** | **124.69** | **0.05** | **703.11** | **0.06** | **97489.28** | **35.09** |
| Improvement | | 27.5 $\times$ | | 77.8% | | 91.2% | | 85.7% | | 83.7% |

Figure 3: **The average primal gap of different methods over 100 instances as the solving process proceeds.** We use Gurobi for implementation and set the time limit to be 1,000 seconds.

**Baselines** We mainly consider two baselines, Predict-and-Search (PS) [18] and Contrastive Predict-and-Search (ConPS) [19]. Contrastive Predict-and-Search (ConPS) is a stronger baseline, leveraging contrastive learning to enhance the performance of PS. For ConPS, we set the ratio of positive to negative samples at ten, using low-quality solutions as negative samples. Our method and baselines can be integrated with conventional solvers such as SCIP [9] and Gurobi [8]. Therefore, we also include SCIP and Gurobi as baselines for a comprehensive comparison. Following Han et al. [18], Gurobi and SCIP are set to focus on finding better primal solutions.

**Metrics** We compare the performance of our method and baselines using the best objective value OBJ within 1,000 seconds. Following the setting in Han et al. [18], we run a single-thread Gurobi for 3,600 seconds and record the best objective value as the best-known solution (BKS) to approximate the optimal value. We calculate the absolute primal gap as the difference between the best objective found by the solvers and the BKS, i.e., $\text{gap}_{\text{abs}} := |\text{OBJ} - \text{BKS}|$. Within the same solving time, a lower absolute primal gap indicates stronger performance. For IS, where most test cases can be solved within the 1,000s, we additionally report the time cost to obtain the optimal value, with a lower time cost indicating a better performance.

**Training and Evaluation** Throughout all the performance experiments, we use *one unified model* for RoME. To train this model, we use a training dataset composed of 720 instances, where the dataset contains 240 IS instances, 240 IP instances and 240 SC instances. The validation set contains 240 instances with 80 IS, IP and SC instances, respectively. The performance of RoME on WA, CA and MIPLIB is the performance of zero-shot generalization. The baselines are trained individually on each dataset using 240 instances for training and 80 instances for validation. For evaluation, we run all the methods on 100 testing instances for each dataset.

## 4.2 Main Results on Cross-Domain Generalization

We first evaluate the effectiveness of RoME trained on multiple domains on various test domains. In this experiment, RoME is trained on IS, IP, and SC. We train just one unified model on these three datasets, and then test it on all five domains, where WA and CA serve as unseen zero-shot testing datasets. The baseline methods are trained on each domain individually and evaluated on the corresponding testing datasets. Table 1 presents the average objective values and $\text{gap}_{\text{abs}}$ achieved by each method using the Gurobi solver across all five combinatorial problem families. Corresponding results for SCIP are provided in Appendix D.2. For the IS dataset, we also report the average solver time in seconds, as all the methods can find the optimal solution within the time limit. The results in Table 1 show that RoME consistently outperforms all baselines across domains. (1) For IS, IP and

SC, RoME is able to achieve the best objective values and absolute primal gap than the baselines trained on the corresponding dataset. This implies that cross-domain training can benefit from the performance of each single domain. (2) Notably, even though not trained on WA and CA, RoME achieves the best performance on these unseen domains, demonstrating its zero-shot generalization capability.

We also report the curves that the average primal gap (defined as $\text{gap}_{\text{rel}} := |OBJ - BKS|/|BKS|$) changes as the solving process proceeds in Figure 3. A rapid decrease in the curves indicates superior solving performance and better convergence. The results in Figure 3 show that RoME exhibits a rapid decline in primal gap and ultimately achieves the lowest primal gap.

Additionally,, we conduct comprehensive ablation studies to investigate the critical components of RoME, evaluating the effects of key elements including DRO, the MoE architecture, embedding perturbation strategies, and expert diversity. Detailed analyses can be found in Appendix D.5.

### 4.3 Zero-Shot Generalization to MIPLIB Problems

To further evaluate the performance of RoME in the real-world complex dataset, we evaluate it on a curated subset of the MIPLIB benchmark [10]. The MIPLIB dataset is a challenging MILP dataset, where the instances are drawn from diverse industrial applications with heterogeneity modeling structures. In this experiment, the baselines are trained and evaluated on splits within MIPLIB. In contrast, RoME is trained on synthetic datasets as used in the main evaluation and directly applied to MIPLIB in a zero-shot fashion. This setting simulates a practical scenario where the solver is deployed in unseen domains without any retraining.

We select two groups of MIPLIB testing instances following the selection criterion in Appendix C.3. The first subset follows Liu et al. [32] and uses the IIS dataset selected from MILPLIB. IIS contains six instances for baselines and five testing instances, where we draw the solving curve in Figure 4. And another subset contains fifteen challenging MIPLIB instances in total, with Figure 5 showcasing solving curves for a representative subset of five selected instances. We find RoME consistently outperforms all the baselines across all the instances, highlighting its strong generalization ability on real-world complex instances. Average results of objective values and $\text{gap}_{\text{abs}}$ across both two selected MIPLIB datasets are presented in Appendix D.1, with each instance result also provided in the same appendix.

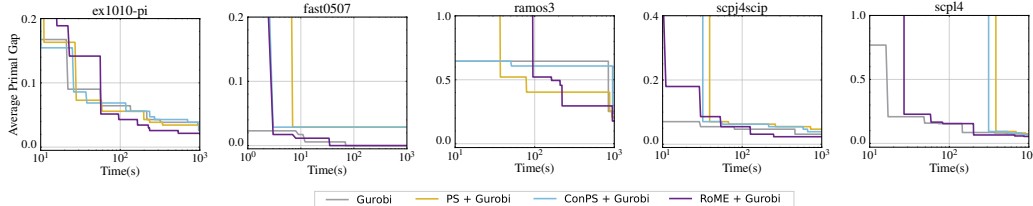

Figure 4: **The primal gap of different methods on five easier instances from IIS dataset as the solving process proceeds.** We use Gurobi for implementation and set the time limit to be 1,000 seconds.

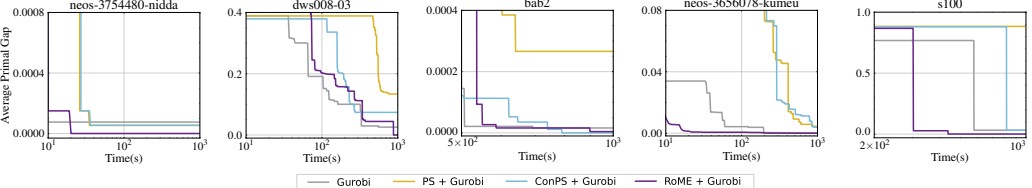

Figure 5: **The primal gap of different methods on five harder instances from MIPLIB as the solving process proceeds.** We use Gurobi for implementation and set the time limit to 1,000s.

### 4.4 Interpretability of RoME

Current works typically adopt GNN-based models, which are inherently treated as black-boxes. However, with the recent progress in large language models [50–52], an increasing number of interpretability techniques have emerged, particularly for MoE architectures, offering new opportunities

to better understand model behavior. Following this, to gain a deeper understanding of RoME, we visualise its expert routing and embedding behaviour. Using the model trained on IS, IP and SC following the main evaluation , we record which expert the gate selects when it solves instances from the five benchmark domains and from five selected challenging instances from MIPLIB, as summarised in Fig.6 (a). We then obtain the task vectors of these instance and plot their distribution through T-SNE, as shown in Fig.6 (b). These interpretability analyses reveal that RoME develops a clear and structured specialization among its experts. Each problem with unique constraint type consistently activates a distinct expert, demonstrating that the mixture model avoids collapse and that individual experts capture unique structural regularities across tasks. The alignment between task embeddings and expert activations shows that RoME learns coherent, task-aware representations: tasks with similar underlying formulations are embedded closely in the latent space and routed to overlapping expert subsets. Furthermore, mixed-constraint MIPLIB instances are positioned near and assigned to the domains they most resemble, explaining RoME's strong generalization and zero-shot adaptability to previously unseen problem families.

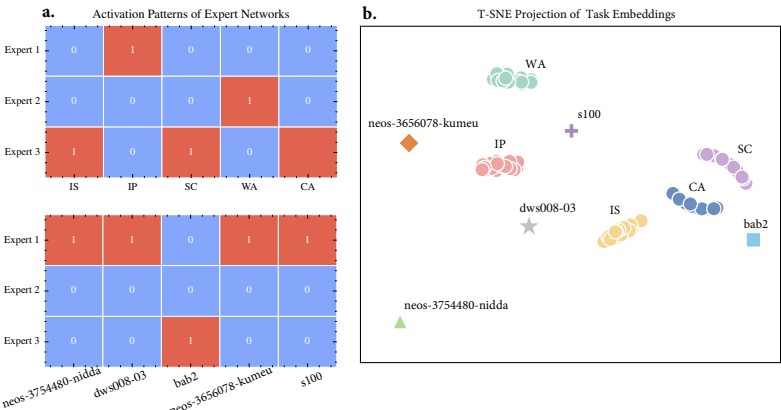

Figure 6: **Model interpretability.** (**a**) Expert network activation patterns across different MILP instances. Red indicates activated experts, while blue denotes inactive ones. (**b**) The distribution of instances in the task embedding space.

## 5  Conclusion

In this paper, we present RoME, a cross-domain MILP solver that pairs a mixture-of-experts architecture with a distributionally robust training objective. The mixture-of-experts architecture enhance representation ability across different domains, while distributionally robust optimization shields against distribution shifts, and together they boost generalization across MILP problem families. Experiments demonstrate substantial improvements over existing learning-based baselines and strong zero-shot performance on challenging MIPLIB instances.

## Acknowledgments and Disclosure of Funding

The authors would like to thank all the anonymous reviewers for their valuable suggestions. This work was supported by the National Natural Science Foundation of China (NSFC, 72571284, 72421002, 62206303, 62273352), the Hunan Provincial Fund for Distinguished Young Scholars (2025JJ20073), the Science and Technology Innovation Program of Hunan Province (2023RC3009), the Foundation Fund Program of National University of Defense Technology (JS24-05), the Major Science and Technology Projects in Changsha, China (kq2301008), the Key Research and Development Program of Hunan Province (2025JK2143), and the Hunan Provincial Innovation Foundation for Postgraduate (XJQY2025044).

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

# A Implementation Details

## A.1 Bipartite Graph Representation

Following previous works [18, 32], we represent each MILP instance as a weighted bipartite graph $\mathcal{G} = (\mathcal{V} \cup \mathcal{W}, \mathcal{E})$, where $\mathcal{V}$ and $\mathcal{W}$ denote the sets of variables and constraints, respectively. The features attached to the bipartite graph are similar to the previous works [18, 32]. The detailed description of these features can be found in Table 2.

Table 2: Description of the variable, Constraint and edge features used in our bipartite graph representation.

| Index | Variable Feature Name | Description |
|---|---|---|
| 0 | Objective | Normalized objective coefficient |
| 1 | Variable coefficient | Average variable coefficient in all constraints |
| 2 | Variable degree | Degree of the variable node in the bipartite graph representation |
| 3 | Maximum variable coefficient | Maximum variable coefficient in all constraints |
| 4 | Minimum variable coefficient | Minimum variable coefficient in all constraints |
| 5 | Variable type | Whether the variable is an integer variable or not) |

| Index | Constraint Feature Name | Description |
|---|---|---|
| 0 | Constraint coefficient | Average of all coefficients in the constraint |
| 1 | Constraint degree | Degree of constraint nodes |
| 2 | Bias | Normalized right-hand-side of the constraint |
| 3 | Sense | The sense of the constraint |

| Index | Constraint Feature Name | Description |
|---|---|---|
| 0 | Coefficient | Constraint coefficient |

## A.2 Implementation Details of the GNN Encoder

In this work, we use a graph neural network (GNN) parameterized by $\theta$ to extract diverse structural representations for MILP instances. For a given instance $\mathcal{I}$, we initialize features for variables $v_i \in \mathbb{R}^6$, constraints $w_j \in \mathbb{R}^4$, and edges $e_{ij} \in \mathbb{R}^1$, then embed them through the MLP layers:

$$h_{v_i}^{(0)} = \text{MLP}_\theta(v_i), \quad h_{w_i}^{(0)} = \text{MLP}_\theta(w_i), \quad h_{e_{ij}}^{(0)} = \text{MLP}_\theta(e_{ij}). \tag{18}$$

Subsequently, we apply two graph convolution layers, each consisting of two interleaved half-convolution operations [11]:

$$
\begin{aligned}
h_{w_i}^{(k+1)} &\leftarrow \text{MLP}_\theta \left( h_{w_i}^{(k)}, \sum_{j:e_{ij} \in \mathcal{E}} \text{MLP}_\phi \left( h_{w_i}^{(k)}, h_{e_{ij}}, h_{v_j}^{(k)} \right) \right) \\
h_{v_i}^{(k+1)} &\leftarrow \text{MLP}_\phi \left( h_{v_i}^{(k)}, \sum_{i:e_{ij} \in \mathcal{E}} \text{MLP}_\theta \left( h_{w_i}^{(k+1)}, h_{e_{ij}}, h_{v_j}^{(k)} \right) \right)
\end{aligned}
\tag{19}
$$

Unlike the previous works that employ extra MLPs for logit prediction, our GNN encoder directly propagates the refined node embeddings to the following modules.

## A.3 Implementation Details of the Baselines

Optimization has broad applications in the real world, including statistical physics [53], complex networks [54, 55], knowledge graph [56, 57], and partial differential equations [58–60]. Recent breakthroughs in computer vision [61–64], graph models [65], and reinforcement learning have

propelled the adoption of machine learning for faster combinatorial optimization solving. In our work, we select two state-of-the-art methods, PS [18] and ConPS [19], as baseline approaches, each exemplifying a prominent direction in recent research. For PS, we utilize the official implementation available at `https://github.com/sribdcn/Predict-and-Search`, and we incorporate the same graph neural network (GNN) encoder from PS into our model architecture. Since ConPS does not have publicly available source code, we meticulously reimplemented its framework based on the original descriptions and performed extensive hyperparameter tuning to replicate its reported performance. All experiments are conducted on a single machine equipped with NVIDIA GeForce RTX 3090 GPUs and AMD EPYC 7402 24-core CPUs running at 2.80GHz. We use Gurobi version 11.0.3 and SCIP version 8.1.0 in all experiments.

For each baseline, we collect 300 instances to serve as training and validation data. Each instance is solved using Gurobi with a single thread for 3600 seconds, during which we record the best 50 solutions. We split the data into 80% for training and 20% for validation, and designate an additional 100 instances as the test set. During training, we set the initial learning rate to 0.0005 and train the model for 10,000 epochs, employing an early stopping mechanism to prevent overfitting. In the testing phase, the parameters $(k_0, k_1, \Delta)$ determine which variables are fixed to 0 or 1 and define the scope of the subsequent search process. These parameters significantly influence the performance of the methods. These parameters used in this work are detailed in Table 3.

Table 3: The partial solution size parameter $(k_0, k_1)$ and neighborhood parameter $\Delta$.

| Benchmark | IS | CA | SC | IP | WA |
|---|---|---|---|---|---|
| PS+Gurobi | (300, 300, 20) | (600,0,1) | (2000,0,100) | (400,5,10) | (0,500,10) |
| ConPS+Gurobi | (1200, 600, 10) | (900,0,50) | (1000,0,200) | (400,5,3) | (0,500,10) |
| RoME+Gurobi | (250, 200, 15) | (350, 0, 55) | (1000, 0, 200) | (60, 35, 55) | (20, 200, 100) |
| PS+SCIP | (300,300, 15) | (400,0,10) | (2000,0,100) | (400,5,1) | (0,600,5) |
| ConPS+SCIP | (1200, 600, 10) | (900,0,50) | (1000,0,200) | (400,5,3) | (0,400,50) |
| RoME+SCIP | (250, 200, 15) | (350, 0, 55) | (1000, 0, 200) | (60, 35, 55) | (20, 200, 100) |

## B  Hyperparameters

The key training parameters are summarized in Table 4.

Table 4: Hyperparameters used in our experiments.

| Name | Value | Description |
|---|---|---|
| embed_size | 128 | The embedding size of the GNN encoder. |
| num_experts | 3 | Number of experts. |
| num_heads | 3 | Number of heads. |
| batch_size | 4 | Number of MILP instances in each training batch. |
| num_epochs | 2000 | Number of max running epochs. |
| lr | 0.0005 | Learning rate for training. |
| perturbation_ratio | 0.1 | Perturbation rate on the task embedding space. |
| expert_diversity_ratio | 0.2 | Expert diversity ratio for training. |

## C  Details on the Benchmarks

### C.1  Benchmarks in Main Evaluation

The CA, SC, and IS benchmark instances are generated following the procedure outlined in [11]. Specifically, the CA instances are created using the algorithm described in [66], while the SC instances are produced based on the method presented in [67]. The IP and WA instances are sourced from the NeurIPS ML4CO 2021 competition [68]. Table 5 provides detailed statistical information for all the instances.

Table 5: Statistical information of the benchmarks we used in this paper.

|  | CA | SC | IP | WA | IS |
|---|---|---|---|---|---|
| Constraint Number | 2593 | 3000 | 195 | 64306 | 5943 |
| Variable Number | 1500 | 5000 | 1083 | 61000 | 1500 |
| Number of Binary Variables | 1500 | 5000 | 1050 | 1000 | 1500 |
| Number of Continuous Variables | 0 | 0 | 33 | 60000 | 0 |
| Number of Integer Variables | 0 | 0 | 0 | 0 | 0 |

## C.2 Benchmarks used for Generalization

To evaluate the generalization capabilities of our methods, we construct larger CA and SC instances using the data generation code from Gasse et al. [11]. The newly generated CA instances contain approximately 2,596 constraints and 4,000 variables on average, while the SC instances comprise 6,000 constraints and 10,000 variables. These problem sizes are substantially larger than those encountered during training, providing a more rigorous evaluation of the models' performance on unseen, large-scale instances.

## C.3 Subset of MIPLIB

Table 6: Statistical information of the instances in the constructed MIPLIB dataset.

| Instance Name | Constraint Number | Variable Number | Nonzero Coefficient Number |
|---|---|---|---|
| ex1010-pi | 1468 | 25200 | 102114 |
| fast0507 | 507 | 63009 | 409349 |
| ramos3 | 2187 | 2187 | 32805 |
| scpj4scip | 1000 | 99947 | 999893 |
| scpk4 | 2000 | 100000 | 1000000 |
| scpl4 | 2000 | 200000 | 2000000 |
| dws008-03 | 16344 | 32280 | 165168 |
| dws008-01 | 6064 | 11096 | 56400 |
| neos2 | 1103 | 2101 | 7326 |
| bab2 | 17245 | 147912 | 2027726 |
| bab5 | 4964 | 21600 | 155520 |
| bab6 | 29904 | 114240 | 1283181 |
| neos-3555904-turama | 146493 | 37461 | 793605 |
| neos-3656078-kumeu | 17656 | 14870 | 59292 |
| supportcase17 | 2108 | 1381 | 5253 |
| fastxgemm-n2r7s4t1 | 6972 | 904 | 22584 |
| neos-3754480-nidda | 203 | 253 | 1488 |
| bg512142 | 1307 | 792 | 3953 |
| tr12-30 | 750 | 1080 | 2508 |
| s100 | 14733 | 364417 | 1777917 |
| s55 | 9892 | 78141 | 317902 |

To evaluate the solvers' performance on challenging real-world instances, we construct two groups of MIPLIB [10] instances. The selection is based on instance similarity, measured using 100 human-designed features as described in [10]. Group 1 follows Liu et al [32], utilizing the IIS dataset from MIPLIB. This dataset comprises 11 instances, divided into a training set of six instances—glass-sc, iis-glass-cov, 5375, 214, 56133, iis-hc-cov, seymour, and v150d30-2hopcds—and a test set of five instances: ex1010-pi, fast0507, ramos3, scpj4scip, and scpk4. We note that baseline methods are trained on this split, whereas our method is evaluated in a zero-shot setting. To further evaluate generalization, we construct Group 2 by selecting five challenging instances from MIPLIB's reported hard instances, dws008-03, bab2, neos-3656078-kumeu, neos-3754480-nidda, and s100, and then

identifying another 10 similar instances based on feature similarity to form the new group. Detailed information on the MIPLIB dataset is provided in Table 6.

## D    More Experiment Results

### D.1    Results on MIPLIB Dataset

To further evaluate the generalization ability of RoME on MIPLIB, we conduct two group experiments. For the IIS subset, the baselines are trained on the designated training instances and then evaluated, whereas RoME is evaluated in a zero-shot setting. And for the remaining instances, every method is run in zero-shot mode. Since the instance sizes vary on different instances, we fix variables by a proportion, using $(k_0, k_1, \Delta) = (0.7, 0, 1000)$. We report two selected MIPLIB dataset from Section 4.3, each containing five selected instances, with the average performance across both datasets for each method presented in Table 7. The detailed MIPLIB results are presented in Table 8.

Table 7: The results in the IIS and the more complicated MIPLIB Instance. We build the ML approaches on Gurobi and set the solving time limit to 3,600s.

|  | IIS (BKS 196.00) | | Hard Instance (BKS -58988.70) | |
| --- | --- | --- | --- | --- |
|  | Obj ↓ | gap$_{abs}$ ↓ | Obj ↓ | gap$_{abs}$ ↓ |
| Gurobi | 211.00 | 15.00 | -58663.30 | 325.40 |
| PS+Gurobi | 210.60 | 14.60 | -57277.70 | 1710.99 |
| ConPS+Gurobi | 210.60 | 14.20 | -58050.10 | 938.58 |
| RoME+Gurobi | **206.80** | **10.80** | **-58988.70** | **0.00** |

Table 8: The best objectives found by the approaches on each test instance in MIPLIB. *BKS* represents the best objectives from the website of MIPLIB `https://miplib.zib.de/index.html`.

|  | BKS | Gurobi | PS+Gurobi | ConPS+Gurobi | RoME+Gurobi |
| --- | --- | --- | --- | --- | --- |
| ex1010-pi | 233.00 | 239.00 | 241.00 | 239.00 | **237.00** |
| fast0507 | 174.00 | 174.00 | 179.00 | 179.00 | **174.00** |
| ramos3 | 186.00 | 233.00 | 225.00 | 225.00 | **224.00** |
| scpj4scip | 128.00 | 132.00 | 133.00 | 133.00 | **131.00** |
| scpl4 | 259.00 | 277.00 | 275.00 | 275.00 | **273.00** |
| dws008-03 | 62831.76 | 64452.67 | 71234.06 | 67473.85 | **62831.75** |
| dws008-01 | 37412.60 | 37412.60 | 39043.26 | 38817.50 | **37415.68** |
| neos2 | 454.86 | 454.86 | 454.86 | 454.86 | **454.86** |
| bab2 | -357544.31 | -357538.58 | -357449.20 | **-357544.31** | -357542.78 |
| bab5 | -106411.84 | -106411.84 | -106411.84 | -106411.84 | **-106411.84** |
| bab6 | -284248.23 | -284248.23 | -284224.56 | -284224.56 | **-284224.56** |
| neos-3555904-turama | -34.7 | -34.7 | -34.7 | -34.7 | **-34.7** |
| neos-3656078-kumeu | -13172.2 | -13171.3 | -13114.0 | -13120.6 | **-13172.2** |
| supportcase17 | 1330.00 | 1330.00 | 1330.00 | 1330.00 | **1330.00** |
| fastxgemm-n2r7s4t1 | 42.00 | 42.00 | 42.00 | 42.00 | **42.00** |
| neos-3754480-nidda | 12939.80 | 12940.78 | 12940.50 | 12940.50 | **12939.80** |
| bg512142 | 184202.75 | 184202.75 | 190193.00 | 190193.00 | **190018.00** |
| tr12-30 | 130596.00 | 130596.00 | 130596.00 | 130596.00 | **130596.00** |
| s100 | -0.1697 | -0.1642 | -0.0198 | -0.1643 | **-0.1696** |
| s55 | -22.15 | -22.15 | -22.15 | -22.15 | **-22.15** |

### D.2    SCIP results in Main Evaluation

Consistent with the experimental setup in Section 4.2, we reported the results of different methods on the SCIP solver as shown in Table 9.

Table 9: Performance comparison across various MILP domains using SCIP solver, under a $1,000$s time limit. We also report the improvement of our method over the traditional solvers in terms of $\text{gap}_{\text{abs}}$.

| | IS (BKS 685.00) | | IP (BKS 11.16) | | SC (BKS 124.64) | | WA (BKS 703.05) | | CA (BKS 97524.37) | |
|---|---|---|---|---|---|---|---|---|---|---|
| | Obj ↑ | Time ↓ | Obj ↓ | $\text{gap}_{\text{abs}}$ ↓ | Obj ↓ | $\text{gap}_{\text{abs}}$ ↓ | Obj ↓ | $\text{gap}_{\text{abs}}$ ↓ | Obj ↑ | $\text{gap}_{\text{abs}}$ ↓ |
| SCIP | 685.00 | 254.63 | 16.55 | 5.39 | 126.43 | 1.38 | 705.77 | 2.72 | 96423.90 | 1100.47 |
| PS+SCIP | 685.00 | 129.35 | 16.18 | 5.02 | 126.65 | 1.60 | 705.21 | 2.16 | 96426.46 | 1097.91 |
| ConPS+SCIP | 685.00 | 78.53 | 16.16 | 5.00 | 126.40 | 1.35 | 705.21 | 2.16 | 96428.83 | 1095.54 |
| RoME+SCIP | **685.00** | **13.55** | **16.08** | **4.92** | **126.27** | **1.22** | **705.08** | **2.03** | **96439.82** | **1084.55** |
| Improvement | | 18.7 × | | 8.7% | | 11.6% | | 25.3% | | 1.4% |

## D.3 Fine-grained Instance-level Results

To provide a fine-grained, instance-level view of RoME's performance, we quantify its performance drop relative to Gurobi. Following standard practice in multi-task learning, we introduce a *win count* metric, the number of test instances, out of 100, on which each method achieves the best objective value. We first compare Gurobi, PS, and RoME on the CA and SC benchmarks. As shown in Table 10, RoME consistently dominates both benchmarks.

Table 10: Total win counts. # denotes the number of instances on which each method achieves the best objective.

| | CA (wins) | SC (wins) |
|---|---|---|
| Gurobi | 26/100 | 5/100 |
| PS+Gurobi | 19/100 | 10/100 |
| **RoME+Gurobi** | **55/100** | **85/100** |

We further separate the instances on which each learning-enhanced solver *wins* or *loses* against Gurobi and report the mean absolute objective gap ($\text{Gap}_{\text{abs}}$) in Table 11. The results show that RoME not only wins on a much larger portion of instances but also yields greater improvements when it wins and smaller degradations when it loses, confirming its stability and overall superiority.

Table 11: Average win/lose gap against Gurobi.

| | CA | SC |
|---|---|---|
| PS+Gurobi win | 178.94 (43/100) | 0.18 (68/100) |
| PS+Gurobi lose | 207.17 (57/100) | 0.41 (32/100) |
| **RoME+Gurobi win** | **295.28 (65/100)** | **0.37 (91/100)** |
| **RoME+Gurobi lose** | **114.93 (35/100)** | **0.10 (9/100)** |

## D.4 Generalization Results

To evaluate the generalization capabilities of our approach, we evaluate its performance on larger instances of the CA and SC problems. These instances, detailed in Appendix C.2, are significantly larger than those used during training. We utilize the model trained on the dataset described in Section 4.1 to perform zero-shot evaluations on these larger instances. The results, presented in Table 12, indicate that our method, RoME, consistently outperforms baseline methods on these larger instances, demonstrating its strong generalization ability.

## D.5 Ablation Studies

**The Effect of DRO and MoE** To clarify the contribution of each component in RoME, we compare the full model with three ablated variants: (i) an MoE-only version trained without the DRO, (ii) a DRO-only version that replaces the MoE with a GCN proposed in the PS [18], and (iii) a single-head variant that removes expert diversity within the MoE architecture. As reported in Table 13, omitting

Table 12: We evaluate the generalization performance on 100 larger instances with a 1,000s time limit.

| | SC (BKS 101.45) | | CA (BKS 115787.97) | |
|---|---|---|---|---|
| | Obj ↓ | gap$_{abs}$ ↓ | Obj ↑ | gap$_{abs}$ ↓ |
| Gurobi | 102.29 | 0.84 | 114960.25 | 827.72 |
| PS+Gurobi | 102.27 | 0.82 | 115228.20 | 559.77 |
| ConPS+Gurobi | 102.18 | 0.73 | 115343.23 | 444.74 |
| RoME+Gurobi | **102.03** | **0.58** | **115787.97** | **0.00** |

either DRO or MoE substantially degrades performance, confirming that these two modules play indispensable roles in RoME's cross-domain generalisation.

Table 13: Impact of DRO and MoE components under a $1,000$s time limit. We report the average best objective values and absolute primal gap.

| | IP (BKS 11.16) | | SC (BKS 124.64) | | WA (BKS 703.05) | | CA (BKS 97524.37) | |
|---|---|---|---|---|---|---|---|---|
| | Obj ↓ | gap$_{abs}$ ↓ | Obj ↓ | gap$_{abs}$ ↓ | Obj ↓ | gap$_{abs}$ ↓ | Obj ↑ | gap$_{abs}$ ↓ |
| Gurobi | 11.43 | 0.27 | 125.21 | 0.57 | 703.47 | 0.42 | 97308.93 | 215.44 |
| MoE-only+Gurobi | 11.27 | 0.11 | 124.65 | 0.01 | 708.33 | 5.18 | 95282.51 | 2241.86 |
| DRO-only+Gurobi | 11.27 | 0.11 | **124.64** | **0.00** | 704.13 | 1.08 | 97194.98 | 329.39 |
| Single-head+Gurobi | **11.18** | **0.02** | 124.83 | 0.19 | 703.14 | 1.09 | 92248.18 | 5276.19 |
| RoME+Gurobi | 11.22 | 0.06 | 124.69 | 0.05 | **703.11** | **0.06** | **97489.28** | **35.09** |

**The Effect of Task Embedding Perturbation**    Furthermore, we evaluate how perturbation influences cross-domain generalization. In RoME, the perturbation is applied in the task-embedding space. We compare this design with three variants: one without any perturbation, one that perturbs the variable-embedding space, and one that perturbs the raw feature space. Table 14 shows that perturbing task embeddings yields the most robust predictions. This aligns with our expectation: the task-embedding space offers a higher-level representation of the entire instance, whereas perturbations in the variable-embedding or feature spaces can distort the underlying problem structure, leading to performance degradation, particularly when transferring across domains.

Table 14: Comparison of different perturbation locations under a $1,000$s time limit. We report the average best objective values and absolute primal gap.

| | IP (BKS 11.16) | | SC (BKS 124.64) | | WA (BKS 703.05) | | CA (BKS 97524.37) | |
|---|---|---|---|---|---|---|---|---|
| | Obj ↓ | gap$_{abs}$ ↓ | Obj ↓ | gap$_{abs}$ ↓ | Obj ↓ | gap$_{abs}$ ↓ | Obj ↑ | gap$_{abs}$ ↓ |
| Gurobi | 11.43 | 0.27 | 125.21 | 0.57 | 703.47 | 0.42 | 97308.93 | 215.44 |
| RoME without Perturb Embedding | 11.61 | 0.45 | 125.27 | 0.63 | 703.20 | 0.15 | 94898.67 | 2625.7 |
| Perturb on Variable Embedding | **11.16** | **0.00** | 124.84 | 0.20 | 703.24 | 0.19 | 94302.71 | 3221.66 |
| Perturb on Variable Features | 11.23 | 0.01 | 124.74 | 0.10 | **703.10** | **0.05** | 91472.19 | 6052.18 |
| RoME+Gurobi | 11.22 | 0.06 | **124.69** | **0.05** | 703.11 | 0.06 | **97489.28** | **35.09** |

We also explore different perturbation magnitudes. As shown in Table 15, a perturbation ratio of 0.1–0.15 yields the best trade-off.

**The Effect of Expert Diversity**    Finally, we study the diversity of experts during training. Table 16 shows that appropriate augmentation of expert diversity improves model performance.

Table 15: Effect of perturbation ratio on CA and SC. The ML approaches are implemented using Gurobi, with a time limit set to 1,000s. '↑' indicates that higher is better, and '↓' indicates that lower is better. We mark the **best values** in bold.

| Perturbation Ratio | SC (BKS 124.64) | | CA (BKS 97524.37) | |
|---|---|---|---|---|
| | Obj ↓ | gap$_{abs}$ ↓ | Obj ↑ | gap$_{abs}$ ↓ |
| 0.05 | 124.70 | 0.06 | 91362.30 | 6162.07 |
| 0.10 | 124.69 | 0.05 | **97489.28** | **35.09** |
| 0.15 | **124.66** | **0.02** | 97374.32 | 150.05 |
| 0.20 | 124.70 | 0.06 | 93362.68 | 4161.69 |

Table 16: Effect of expert diversity on CA and SC. The ML approaches are implemented using Gurobi, with a time limit set to 1,000s. '↑' indicates that higher is better, and '↓' indicates that lower is better. We mark the **best values** in bold.

| Expert Diversity Ratio | SC (BKS 124.64) | | CA (BKS 97524.37) | |
|---|---|---|---|---|
| | Obj ↓ | gap$_{abs}$ ↓ | Obj ↑ | gap$_{abs}$ ↓ |
| 0.1 | 125.27 | 0.63 | 94898.67 | 2625.70 |
| 0.2 | **124.69** | **0.05** | **97489.28** | **35.09** |

# E    Discussions

## E.1    Limitations

**Higher training cost.** Although cross-domain solvers demonstrate strong performance, their training requires substantial diverse data and incurs relatively high computational costs. Limited by GPU memory capacity, we selected 300 samples per cross-domain dataset. Our observations indicate that while model performance improves with larger training sets, this comes at the cost of significantly increased computational overhead.

**Waiting for more sophisticated designs.** Cross-domain MILP solving remains in its early stage and currently lacks sophisticated architectural designs. For instance, the field would benefit from more efficient sparse MoE architectures specifically optimized for large-scale MILP applications.

## E.2    Future works

**Better optimization algorithms.** In this work, we employ DRO to balance training across different domains, while simulating cross-domain distribution shifts through perturbations in the task embedding space. We note that more sophisticated mathematical techniques could further enhance model robustness by explicitly incorporating uncertainty. Future research directions include developing advanced optimization algorithms to strengthen robustness and improve cross-domain generalization performance.

**Better model architecture.** Future research directions may explore more sophisticated MoE architectures to enhance the current framework, such as incorporating sparse MoE variants or attention mechanisms. Additionally, replacing the current GNN encoder with more expressive graph neural network models could further improve representation learning capabilities.

## E.3    Broader Impacts

This paper presents RoME, a domain-robust Mixture-of-Experts framework that predicts high-quality initial solutions for diverse MILP problems without retraining. By plugging RoME into off-the-shelf solvers, practitioners in logistics, network design, and manufacturing can accelerate solution times and lower computational costs. Because RoME learns structural patterns rather than proprietary coefficients, it can be safely deployed on sensitive industrial instances while easing data-collection burdens and expanding access to learning-based optimization where labeled solutions are scarce.

