# OpenReview forum: "RoME: Domain-Robust Mixture-of-Experts for MILP Solution Prediction across Domains"
_NeurIPS.cc/2025/Conference — NeurIPS 2025 poster_

### Official Review · Reviewer_i5gC · 2025-07-01

**Clarity:** 3
**Significance:** 2
**Originality:** 2
**Rating:** 3
**Confidence:** 4

**Summary:**

The paper introduces a machine learning model designed to predict solutions for mixed-integer linear programs (MILPs) to serve as warm starts. Specifically, the model is a variant of the mixture-of-experts framework. The predicted solution is then used to initialize the solver, which subsequently solves the original MILP constrained to a neighborhood around the warm-start solution.

**Questions:**

- Question 1: What is the denominator in equation (2) summation over (x')? My concern is the feasible region of a MIP can have uncountably infinite solution.

- Question 2: The paper seems to claim its method works on general MIP at first, but starting from line 123, it started to assume that MIP only contains binary variables. Can you clarify this point?

- Question 3: What is your intuition for the "fully factorized solution" assumption in line 124? Besides the part that it makes it easy to design a loss function, is there any mathematical justification for that? Ignoring the part the definition of p(x|I) is not yet well-defined, it is doubtful that this assumption can be true, as given an instance I, the value of x_1 can change the feasible region of x_2. In addition, using this assumption, are we ignoring the continuous part of the solution? it seems like even if the continuous part give bad objective, we still assign high probability.

- Question 4: The definition of x^(i)_j in line 129 is a bit confusing. Why there is no j in the right-hand side?

- Question 5: In Append A.3 , It is said the the selection of k_0, k_1 and Delta have great influences. Given that in Table 3, different methods use different configuration of these values. Would you be able to provide tables/plots showing the impact of these values when varying?

- Question 6: I do not understand why, in Figure 3, the lines plot do not start second 0, but rather at the 10th second (4 out of 5)? Is it because the solver takes a while to find its first feasible solution or something else?

- Question 7: I am curious about the quality of the binary solution returned by RoME. I think it give us more insight if you can show the objective of the returned solution of RoME or how many of it was feasible (for all instances in your experiments). For the non-pure binary case, we can assign value for the continuous variables by the solving the LP when fixed all binary variables at the predicted solution.

**Ethical Concerns:**

["NO or VERY MINOR ethics concerns only"]

**Final Justification:**

I have given my response in the comment sections. (Sorry I forgot to raise my rating)

**Limitations:**

Yes

**Paper Formatting Concerns:**

I do not notice any major formatting issues.

**Quality:**

2

**Strengths And Weaknesses:**

**Strength**

- The paper provides an interesting framework (RoME) for the the prediction of MIP.

**Weakness**

- The theoretical foundation for the choice of the loss function used in training seems weak (see Questions).

- As the main contribution of the paper is the RoME model, there should also be numerical study for the solution returned by the model, not just a combined effect of the model and solving a constrained MILP (see Questions)

I am willing to re-evaluate my rating if my concerns are answered.

---

> ### Author Rebuttal · Authors · 2025-07-31
>
> Dear Reviewer i5gC,
>
> Thank you for your insightful and valuable comments. We sincerely hope our rebuttal adequately address your concerns. If so, we would deeply appreciate it if you could consider raising your score. If not, please let us know your further concerns, and we will continue actively responding to your comments.
>
> **Q1: The denominator in equation (2).**
>
> > What is the denominator in equation (2) summation over (x')? My concern is the feasible region of a MIP can have uncountably infinite solution.
>
> Thank you for your feedback! We would like to clarify Equation (2) as follows.
>
> - First, it's important to note that the denominator in Equation (2) does not represent an infinite sum of solutions. In our implementation, we utilize Gurobi to run training instances for 3,600 seconds, resulting in the **50 solutions with the highest objective values**. The denominator, therefore, is a summation over these top 50 solutions.
> - More specifically, following the previous works [2-3], RoME also aims to **approximate high-quality solution distributions**. This means we focus solely on solutions with favorable objective values during training. Consequently, we use the top 50 solutions to estimate these high-quality distributions. As you noted, the total solution space can be infinite, making a summation over the entire space impractical.
>
> **Q2: The clarification of our methods on general MILPs.**
>
> > The paper seems to claim its method works on general MIP at first, but starting from line 123, it started to assume that MIP only contains binary variables. Can you clarify this point?
>
> Thank you for your insightful comments! To address your concerns, we have conducted experiments on general MILPs that include integer, binary, and continuous variables.
>
> - **Our method is applicable to general MILPs**. In Section 6.1.3 of [1], the authors indicate that the model can be generalized to non-binary variables with certain modifications. Following the guidance in [1], we have adapted our model to accommodate general MILPs.
> - We performed experiments on four challenging instances from MIPLIB, which are general MILP problems featuring integer variables. The results, presented in Table 1, demonstrate our method's superior performance compared to the baseline approaches. As in the main-text experiments, we first select a small set of similar instances (based on the instance-similarity metrics) and use them to train the PS model, which is then evaluated on the testing instances. By contrast, our method is applied to the same testing instances **in zero-shot mode**, requiring no additional training.
>
> Table 1: Experiments on the general MILP instances.
>
> |                    | #Integer | #Binary | #Continuous | BKS         | PS+Gurobi   | RoME+Gurobi |
> | ------------------ | -------- | ------- | ----------- | ----------- | ----------- | ----------- |
> | neos-3656078-kumeu | 4455     | 9755    | 660         | -13172.2    | -13114.0    | **-13172.2**    |
> | buildingenergy     | 26287    | 0       | 128691      | 33283.85    | 33284.03    | **33283.85**    |
> | germanrr           | 5251     | 5323    | 239         | 47095869.64 | 47146028.22 | **47135773.34** |
> | adult-max5features | 41       | 32597   | 36          | 5642.12     | 5679.12     | **5642.12**     |
>
> **Q3: Explanation on the "fully factorized solution" assumption.**
>
> > What is your intuition for the "fully factorized solution" assumption in line 124? Besides the part that it makes it easy to design a loss function, is there any mathematical justification for that? Ignoring the part the definition of p(x|I) is not yet well-defined, it is doubtful that this assumption can be true, as given an instance I, the value of x_1 can change the feasible region of x_2. In addition, using this assumption, are we ignoring the continuous part of the solution? it seems like even if the continuous part give bad objective, we still assign high probability.
>
> Thanks for your question!  We would like to address your points regarding the definition of our probability distribution and the justification for the independence assumption.
>
> - **Clarification of the definition of the probability distribution $p(x|\mathcal{I})$**. This is not a theoretical distribution over all possible solutions, but rather an **empirical probability** derived from a collected set of high-quality feasible solutions for a given instance $\mathcal{I}$. To generate the training data for an instance, we first solve it to collect a set of top-performing solutions (e.g., the best 50 solutions found within a time limit). The probability for a single binary variable, $p_\theta(x_i=1|\mathcal{I})$, is then computed as the frequency of that variable taking the value 1 across this set of elite solutions. In essence, it is the average value of that variable in the collected solution set, weighted by solution quality.
> - **The Assumption is widely adopted in the previous works**. We acknowledge that variable independence is a strong assumption that may not hold true in practice. However, this is a widely adopted simplification in the field for reasons of computational tractability, as seen in related works [1-3]. The core intuition is that if a variable takes on a consistent value (e.g., 1) across the vast majority of top-tier solutions, then its marginal probability will be high.
> - **The variable dependencies are captured to a degree by the GNN encoder**. The GNN processes the entire MILP instance as a single graph, and through its multi-layer message-passing mechanism, the final embedding for each variable is informed by the global problem structure, including its relationships with other variables and constraints. Therefore, the complex relationships between variables are considered during the feature extraction stage, even if they are not explicitly modeled in the final probabilistic output.
> - **Explanation on the continuous variables**. The computational complexity of solving MILPs stems primarily from the combinatorial nature of the **integer** and **binary variables**. By predicting the values for a significant portion of the integer and binary variables, we effectively prune the search tree and guide the solver to a much smaller, more promising region of the solution space. Finding the optimal values for the remaining variables and continuous variables becomes significantly more tractable for the solver.
>
> **Q4 Typos**
>
> > The definition of x^(i)_j in line 129 is a bit confusing. Why there is no j in the right-hand side?
>
> Thank you for pointing this typo.  We have corrected it to $x_j^{(i)} = p_{\theta}(x_j^{(i)}=1|\mathcal{I})$.
>
> **Q5 Sensitivity of $k_0$, $k_1$ and $\Delta$**
>
> > In Append A.3 , It is said the the selection of k_0, k_1 and Delta have great influences. Given that in Table 3, different methods use different configuration of these values. Would you be able to provide tables/plots showing the impact of these values when varying?
>
> Thank you for the insightful comments. To address them, we have conducted an additional sensitivity study on the CA benchmark. The results are summarized in Table 2. Please note that PS is trained directly on the CA dataset, whereas RoME has never seen CA during training. Despite this disadvantage, RoME still outperforms PS across all parameter settings.
>
> Table 2: Sensitivity of hyperparameters.
>
> | $k_0$ | $k_1$ | $\Delta$ | PS+Gurobi | RoME+Gurobi |
> | ----- | ----- | -------- | --------- | ----------- |
> | 400   | 0     | 60       | 97280.70  | **97489.28**    |
> | 400   | 10    | 60       | 97235.69  | **97273.60**    |
> | 400   | 20    | 60       | 97224.88  | **97272.06**    |
> | 450   | 0     | 40       | 97231.42  | **97265.50**    |
> | 450   | 20    | 60       | 97221.65  | **97228.26**    |
> | 500   | 0     | 60       | 97194.68  | **97228.06**    |
> | 500   | 10    | 40       | 97188.40  | **97227.41**    |
> | 500   | 10    | 60       | 97195.65  | **97220.95**    |
>
> **Q6 Explanation on the Figure 3**
>
> > I do not understand why, in Figure 3, the lines plot do not start second 0, but rather at the 10th second (4 out of 5)? Is it because the solver takes a while to find its first feasible solution or something else?
>
> Thanks for your feedback! We observed that our approach usually finds the first feasible solution at around 10th second—except on the IS benchmark, which is relatively easy, so all methods locate a feasible point almost instantly. This observation matches prior works [2-4], which also plot results starting from the 10th second.
>
> **Q7 Quality of RoME**
>
> > I am curious about the quality of the binary solution returned by RoME. I think it give us more insight if you can show the objective of the returned solution of RoME or how many of it was feasible (for all instances in your experiments). For the non-pure binary case, we can assign value for the continuous variables by the solving the LP when fixed all binary variables at the predicted solution.
>
> Thank you for the comments! Methods in the **Predict-and-Search** family [2-3] learn a distribution of high-quality solutions and then obtain a **partial** assignment of variables. Following your suggestion, we fixed these predicted variables, solved the remaining problem, and reported the resulting objectives for PS and RoME on the SC and WA benchmarks in Table 3.
>
> Table 3: Results of directly fixing variables.
>
> |             | SC BKS: 124.64 | WA BKS: 703.05 |
> | ----------- | -------------- | -------------- |
> | PS+Gurobi   | 125.41         | 703.80         |
> | RoME+Gurobi | **125.08**         | **703.65**         |
>
> [1] Solving Mixed Integer Programs Using Neural Networks.
>
> [2] A GNN-Guided Predict-and-Search Framework for Mixed-Integer Linear Programming. ICLR 2023.
>
> [3] Contrastive Predict-and-Search for Mixed Integer Linear Programs. ICML 2024.
>
> [4] Differentiable Integer Linear Programming. ICLR 2025.

---

> > ### Comment · Reviewer_i5gC · 2025-08-06
> > **The author has addressed all of my questions. However, it wasn't enough  to raise my evaluation.**
> >
> > First of all, thanks for indulging me in answering all of my question!
> >
> > - Please ensure that you make edit to the typos, clarification in the definition and assumption. In addition, please also add citations for the variable dependence assumption you have, as you pointed out in your answer.
> >
> > - Your answer on Q6 makes it more confusing to me. Isn't the point of RoME is to predict a good solution initially? So why did it take a while to obtain an optimal solution?
> >
> > - In the new numerical studies you provided, why did you only provide on 1 or 2 classes of problem instead of all 5 in the paper?

---

> > > ### Author Response · Authors · 2025-08-07
> > > **Thank you for your kind support and reponse to your further comments.**
> > >
> > > Dear Reviewer i5gC,
> > >
> > > Thank you for your insightful and valuable comments. Below, we address your further comments in detail.
> > >
> > > > Please ensure that you make edit to the typos, clarification in the definition and assumption. In addition, please also add citations for the variable dependence assumption you have, as you pointed out in your answer.
> > >
> > > Thanks for your valuable comments! Based on the previous discussions, we have edited the relevant parts of the paper to clarify the probability definition, justify the factorization assumption with citations, and correct typos in the main paper.
> > >
> > > > Your answer on Q6 makes it more confusing to me. Isn't the point of RoME is to predict a good solution initially? So why did it take a while to obtain an optimal solution?
> > >
> > > Thank you for your comments! Your interpretation of RoME's ability to predict a good initial solution is correct; however, it's important to clarify that this solution is only a partial one.
> > >
> > > - To construct the partial solution, RoME predicts the marginal probabilities $p_\theta(x_i=1|\mathcal{I})$ for the binary variables within the MILP instance. Based on these probabilities, we select the variables with the highest confidence: the top $k_1$ variables are fixed at a value of 1, while the bottom $k_0$ variables are fixed at a value of 0. This helps us construct a partial solution.
> > > - Using this partial solution, the solver then searches for the remaining unfixed variables to identify a feasible solution, and subsequently the optimal solution, based on the predicted partial solution. Due to the presence of unfixed variables, we cannot define the objective values for the partial solution. Consequently, as illustrated in Figure 3, the line plot begins at second 10, as the solver was able to find the first feasible solution at this time.
> > >
> > > > In the new numerical studies you provided, why did you only provide on 1 or 2 classes of problem instead of all 5 in the paper?
> > >
> > > Thank you for your comment! First, we apologize for initially including results on only 1 or 2 problems. We would like to clarify that conducting extensive experiments across all five datasets within a short rebuttal period involves **a huge amount of work**. Since our paper mainly focuses on cross-domain generalization, we initially conducted experiments on out-of-domain datasets to highlight this capability. However, to better address your concerns, we have now **extended the experiments** to cover other datasets.
> > >
> > > - For Q5, we add sensitivity analyses on $k_0$, $k_1$, and $\Delta$ for the IP, IS, SC, and WA datasets. The results are shown in the following tables.
> > >
> > >   Table 1: Sensitivity analysis for SC.
> > >
> > >   | k0   | k1   | delta | PS+Gurobi | RoME+Gurobi |
> > >   | ---- | ---- | ----- | --------- | ----------- |
> > >   | 1000 | 0    | 200   | 125.17    | **124.69**  |
> > >   | 1000 | 0    | 150   | 126.11    | **125.28**  |
> > >   | 1200 | 0    | 200   | 126.06    | **125.22**  |
> > >   | 1000 | 10   | 200   | 126.07    | **125.24**  |
> > >
> > >   Table 2: Sensitivity analysis for IP.
> > >
> > >   | k0   | k1   | delta | PS+Gurobi | RoME+Gurobi |
> > >   | ---- | ---- | ----- | --------- | ----------- |
> > >   | 60   | 35   | 55    | 11.40     | **11.22**   |
> > >   | 60   | 35   | 45    | 12.33     | **11.89**   |
> > >   | 70   | 35   | 55    | 12.20     | **11.63**   |
> > >   | 60   | 45   | 55    | 12.33     | **11.82**   |
> > >
> > >   Table 3: Sensitivity analysis for IS.
> > >
> > >   | k0   | k1   | delta | PS+Gurobi | RoME+Gurobi |
> > >   | ---- | ---- | ----- | --------- | ----------- |
> > >   | 250  | 200  | 15    | 685.00    | **685.00**  |
> > >   | 250  | 200  | 10    | 684.25    | **684.75**  |
> > >   | 260  | 200  | 15    | 684.25    | **685.00**  |
> > >   | 250  | 210  | 15    | 684.25    | **685.00**  |
> > >
> > >   Table 4: Sensitivity analysis for WA.
> > >
> > >   | k0   | k1   | delta | PS+Gurobi | RoME+Gurobi |
> > >   | ---- | ---- | ----- | --------- | ----------- |
> > >   | 20   | 200  | 100   | 703.47    | **703.11**  |
> > >   | 20   | 200  | 80    | 703.49    | **703.31**  |
> > >   | 25   | 200  | 100   | 703.49    | **703.38**  |
> > >   | 20   | 220  | 100   | 703.48    | **703.30**  |
> > >
> > > - For **Q7**, we conduct additional experiments on the IP, IS, and CA benchmarks. The results are reported below.
> > >
> > >   Table 5: Results of directly fixing variables on IP, IS and CA benchmarks.
> > >
> > >   |             | IP BKS: 11.16 | IS 685     | CA 97524.37  |
> > >   | ----------- | ------------- | ---------- | ------------ |
> > >   | PS+Gurobi   | 11.43         | 683.66     | 96123.31     |
> > >   | RoME+Gurobi | **11.40**     | **683.75** | **96218.23** |
> > >
> > > We sincerely hope that we have addressed all of your concerns. We would greatly appreciate your feedback to help us understand whether our responses have fully resolved your questions. If so, we would deeply appreciate it **if you could consider raising your score**. If not, please let us know your further concerns, and we will continue actively responding to your comments. We sincerely thank you once more for your insightful comments and kind support.

---

> > > > ### Author Response · Authors · 2025-08-08
> > > >
> > > > Dear Reviewer i5gC,
> > > >
> > > > We would like to extend our sincere gratitude for the time and effort you have devoted to reviewing our submission. Your valuable feedback, insightful comments, and constructive suggestions have been invaluable to us, guiding us in improving the quality of our work!
> > > >
> > > > We eagerly await your feedback to understand if our responses have adequately addressed all your concerns. If so, we would deeply appreciate it if you could raise your score. If not, we are eager to address any additional queries you might have, which will enable us to enhance our work further.
> > > >
> > > > Once again, thank you for your guidance and support.
> > > >
> > > > Best,
> > > >
> > > > Authors

---

> ### Author Response · Authors · 2025-08-09
>
> Dear Reviewer i5gc,
>
> We sincerely thank you for your time and efforts during the rebuttal process. We are writing to gently remind you that the author-reviewer discussion period will end in less than 10 hours. We have responded to your further comments and eagerly await your feedback, and we sincerely hope that our response has properly addressed your concerns. We would deeply appreciate it if you could kindly point out your further concerns so that we could keep improving our work. We sincerely thank you once more for your insightful comments and kind support.
>
> Best,
>
> Authors

---

### Official Review · Reviewer_4Hsn · 2025-07-02

**Clarity:** 3
**Significance:** 2
**Originality:** 3
**Rating:** 4
**Confidence:** 3

**Summary:**

The paper addresses the problem of domain generalization for MILP solution prediction using machine learning. The author proposes a general-purpose framework that can perform well across different MILP problem distributions,which leverages a Mixture-of-Experts model that dynamically selects experts based on learned task embeddings. Training is guided by a novel two-level DRO objective that mitigates domain shifts and promotes robustness. Results show that a single RoME model trained on multiple domains generalizes better than single-domain models.

**Questions:**

1. Although the model performs well on average, for some problem, a performance drop can be observed compared to Gurobi. This performance drop should be quantified and reported.
2. The effect of the number of experts in the MoE architecture is not explored, which is essential to understand the scalability and architectural sensitivity of the proposed approach.

**Ethical Concerns:**

["NO or VERY MINOR ethics concerns only"]

**Final Justification:**

I've read the rebuttal. The rebuttal addresses my concerns. I will maintain the current scores.

**Limitations:**

Yes

**Quality:**

3

**Strengths And Weaknesses:**

#Strengths
- This paper is well-written and easy to understand
- This paper is technically sound
- The experiments are extensive

#Weakness
- Although the model performs well on average, for some problem, a performance drop can be observed compared to Gurobi. This performance drop should be quantified and reported.
- The effect of the number of experts in the MoE architecture is not explored, which is essential to understand the scalability and architectural sensitivity of the proposed approach.
- The paper includes minor inconsistencies and typos (e.g., “T-SNE” should be “t-SNE” with lowercase “t”).

---

> ### Author Rebuttal · Authors · 2025-07-31
>
> Dear Reviewer 4Hsn,
>
> Thank you for your insightful and valuable comments. We sincerely hope our rebuttal adequately address your concerns. If so, we would deeply appreciate it if you could consider raising your score. If not, please let us know your further concerns, and we will continue actively responding to your comments.
>
> **W1 & Q1 Need to quantify performance drop vs. Gurobi**
>
> > Although the model performs well on average, for some problem, a performance drop can be observed compared to Gurobi. This performance drop should be quantified and reported.
>
> Thank you for highlighting the importance of quantifying the performance drop against Gurobi. Following standard practice in multi-task learning, we introduce a “**win**” metric—i.e., the number of test instances (out of 100) on which each method attains the best objective. First, we compare Gurobi , PS, and RoME together on the CA and SC benchmarks, as shown in Table 1.
>
> Table 1: The total win counts of each method.
>
> |                 | CA (wins)  | SC (wins)  |
> | --------------- | ---------- | ---------- |
> | Gurobi          | 26/100     | 5/100      |
> | PS+Gurobi       | 19/100     | 10/100     |
> | RoME+Gurobi | **55/100** | **85/100** |
>
> RoME dominates on both benchmarks. We then separate the instances on which each learning-enhanced solver **wins** or **loses** against Gurobi and report the mean absolute objective gap ($\text{Gap}_{\text{abs}}$) in Table 2.
>
> Table 2: The win/lose gap against Gurobi of each method.
>
> |                      | CA                  | SC                |
> | -------------------- | ------------------- | ----------------- |
> | PS+Gurobi win        | 178.94 (43/100)     | 0.18 (68/100)     |
> | PS+Gurobi lose       | 207.17 (57/100)     | 0.41 (32/100)     |
> | RoME+Gurobi win  | **295.28 (65/100)** | **0.37 (91/100)** |
> | RoME+Gurobi lose | **114.93 (35/100)** | **0.10 (9/100)**  |
>
> RoME not only wins far more often but also gains more when it wins and loses less when it falls short. For a concrete view, the Table 3 lists the objective values on the first 20 CA instances:
>
> Table 3: The objective of partial instances obtained by each method.
>
> |      | Gurobi       | PS+Gurobi | RoME+Gurobi  |
> | ---- | ------------ | --------- | ------------ |
> | 1    | 97078.81     | 97064.76  | **97337.83** |
> | 2    | 98538.49     | 99177.49  | **99484.99** |
> | 3    | 98290.87     | 97450.07  | **97590.69** |
> | 4    | **95981.26** | 95614.13  | 95966.05     |
> | 5    | **98260.07** | 98528.65  | 97705.04     |
> | 6    | 97524.37     | 96509.32  | **97524.37** |
> | 7    | **99969.14** | 99753.87  | 99887.29     |
> | 8    | **98248.98** | 98212.79  | 97823.80     |
> | 9    | **97644.47** | 97640.97  | 97553.09     |
> | 10   | **99367.75** | 99028.51  | 98837.78     |
> | 11   | 99538.28     | 99147.39  | **99608.86** |
> | 12   | **93831.97** | 93668.45  | 93551.63     |
> | 13   | 95768.25     | 95984.20  | **96180.39** |
> | 14   | **98510.61** | 97720.41  | 97598.75     |
> | 15   | 99458.79     | 98479.86  | **99860.52** |
> | 16   | 95993.82     | 96851.43  | **96851.43** |
> | 17   | 98138.46     | 97219.51  | **98735.76** |
> | 18   | 97137.85     | 96635.77  | **97469.91** |
> | 19   | **96691.47** | 96087.46  | 96480.80     |
> | 20   | **97592.81** | 97062.06  | 97125.03     |
>
> **W2 & Q2 Effect of expert count on scalability unexplored**
>
> > The effect of the number of experts in the MoE architecture is not explored, which is essential to understand the scalability and architectural sensitivity of the proposed approach.
>
> Thank you for stressing this point. As **Fig. 1(b) in the main text** shows, scaling the number of experts together with the number of training tasks **consistently improves performance**. We further evaluate scalability by varying the expert count on four benchmarks (For each benchmark, we select 20 instances) in Table 4:
>
> Table 4: Effect of expert count on RoME performance.
>
> | # Experts | IP BKS: 13.78 | WA BKS: 724.19 | SC BKS: 121.60 | CA BKS: 97428.41 |
> | --------- | ------------- | -------------- | -------------- | ---------------- |
> | 1         | 14.54         | 724.69         | 121.99         | 90769.83         |
> | 2         | 14.81         | 730.69         | 122.39         | 94101.89         |
> | 3     | **13.85**     | **724.25**     | **121.60**     | **97463.46**     |
>
> **W3 Typos**
>
> > The paper includes minor inconsistencies and typos (e.g., “T-SNE” should be “t-SNE” with lowercase “t”).
>
> Thank you. We have corrected these typos in the manuscript.

---

> > ### Comment · Reviewer_4Hsn · 2025-08-04
> >
> > I've read the rebuttal. I thank the author for the clarification and additional experiments, which address my concerns. I will maintain the current scores.

---

> > > ### Author Response · Authors · 2025-08-04
> > > **Thank you for your kind support.**
> > >
> > > Dear Reviewer 4Hsn,
> > >
> > > Thanks for your kind support and for helping us improve the paper. We sincerely appreciate your valuable suggestions.
> > >
> > > Best,
> > >
> > > Authors

---

### Official Review · Reviewer_SKRg · 2025-07-02

**Clarity:** 3
**Significance:** 4
**Originality:** 3
**Rating:** 4
**Confidence:** 3

**Summary:**

This paper proposes a domain-Robust Mixture-of-Experts (RoME) framework for computing high-quality initial solutions to mixed-integer linear programs (MILPs) that can generalize to different types of MILPs. The framework leverages a two-stage MoE architecture (on both the encoder and decoder parts). In terms of the intra-domain generalization, the authors propose a fuzzy consistency loss that forces the routing results of the original and perturbed task embeddings to be the same, which can potentially make it generalize to unseen types of problems. For the inter-domain generalization, the authors use group-DRO that puts more focus on the hardest domain in the training mini-batch.

The experiment's results look promising. One RoME model combined with Gurobi can either largely reduce Gurobi's running time or improve the quality of the solutions.

**Questions:**

1. See cons above.
2. In terms of the testing problem sizes, what's the gap between them and the real-world problem size?

**Ethical Concerns:**

["NO or VERY MINOR ethics concerns only"]

**Final Justification:**

All my questions are resolved, and I will keep my score unchanged.

**Quality:**

3

**Strengths And Weaknesses:**

Pros:

1. The entire RoME framework shows a good generalization ability to unseen types of problems. All the proposed gadgets contribute to the performance.
2. The paper is well written and very easy to follow.

Cons:

1. The reason why the fuzzy consistency loss works seems unclear. In the high-dimensional task embedding space, why can forcing the original and perturbed outputs to be the same help generalize? Do you have more insights into this part?

---

> ### Author Rebuttal · Authors · 2025-07-31
>
> Dear Reviewer SKRg,
>
> Thank you for your insightful and valuable comments. We sincerely hope our rebuttal adequately address your concerns. If so, we would deeply appreciate it if you could consider raising your score. If not, please let us know your further concerns, and we will continue actively responding to your comments.
>
> **W1 & Q1 Rationale of fuzzy consistency loss remains unclear**
>
> > The reason why the fuzzy consistency loss works seems unclear. In the high-dimensional task embedding space, why can forcing the original and perturbed outputs to be the same help generalize? Do you have more insights into this part?
>
> Thank you for raising this point! To clarify the effect of the fuzzy-consistency loss, we have conducted an ablation study in which we **removed** this loss—that is, we no longer require the outputs of the original and perturbed task embeddings to be the same. We evaluate this change on the **Set Cover** benchmark and on the MIPLIB instance **`bab2`** (a real-world instance which also contains the `set_covering` constraint type). The results are shown Table 1.
>
> Table 1: Ablation study of fuzzy consistency loss.
>
> |      | PS + Gurobi | RoME w/o $\mathcal{L}_{robust}$ + Gurobi | RoME + Gurobi  |
> | ---- | ----------- | ---------------------------------------- | -------------- |
> | SC   | 125.17      | 125.17                                   | **124.69**     |
> | bab2 | -357449.20  | -357168.59                               | **-357542.78** |
>
> The improvement confirms the effectiveness of the fuzzy-consistency loss. We attribute the gains to three factors:
>
> - **More accurate routing.** A task embedding represents the structural pattern of a MILP instance and is directly consumed by the MoE gate. The loss encourages ***similar*** embeddings to select the ***same*** expert subset. In Section 4.4 we observed that both SC and bab2 activate **Expert #3**. When the loss is removed, SC still selects **Expert #3**, but bab2 mistakenly switches to **Expert #2**, degrading performance.
> - **Alignment with cross-domain reducibility.** From a computational-complexity viewpoint, most MILP classes are NP-complete, implying that different MILPs can, in principle, be reduced to one another—a fact confirmed by recent advances in graph-based combinatorial optimization [1]. Building on this insight, we exploit structural similarities (e.g., the IS and CA domains both have the `set_packing` constraints) so that related domains route to the same expert subset, enhancing generalization. Perturbing task embeddings partially simulates cross-domain shifts; forcing the perturbed and original embeddings to yield identical outputs tightens this alignment and further boosts generalization.
> - **Enhanced robustness.** From an adversarial-training viewpoint, embedding perturbations stabilize the training of DRO, helping RoME capture high-level patterns that persist across domains.
>
> [1] UniCO: On Unified Combinatorial Optimization via Problem Reduction to Matrix-Encoded General TSP. ICLR 2025.
>
> **Q2 Gap between benchmark and real-world problem sizes**
>
> > In terms of the testing problem sizes, what's the gap between them and the real-world problem size?
>
> Thank you for your feedback! **Appendix C** lists the sizes of all test instances, covering both our synthetic benchmarks and the real-world **MIPLIB** benchmarks. MIPLIB instances originate from **industrial production, transportation, and other operational settings**. And their variable and constraint counts can reach **hundreds of thousands**. Here we select some of them which are reported in Table 2, and the complete information is detailed in **Appendix C**.
>
> Table 2: Statistical information of some partial benchmarks.
>
> | Benchmark           | #Constraint | #Variable |
> | ------------------- | ----------- | --------- |
> | CA                  | 2593        | 1500      |
> | SC                  | 3000        | 5000      |
> | WA                  | 64306       | 61000     |
> | scpl4               | 2000        | 200000    |
> | bab6                | 29904       | 114240    |
> | neos-3555904-turama | 146493      | 37461     |
> | s100                | 14733       | 364417    |

---

### Official Review · Reviewer_SAaz · 2025-07-06

**Clarity:** 3
**Significance:** 3
**Originality:** 3
**Rating:** 5
**Confidence:** 4

**Summary:**

- The paper introduces RoME, a domain-robust Mixture-of-Experts framework designed to predict solutions for MILP problems across various domains. Traditional MILP solvers struggle with generalization across different problem distributions, but RoME addresses this by dynamically routing problem instances to specialized experts based on learned task embeddings.

- RoME utilizes a Mixture-of-Experts (MoE) architecture combined with a two-level Distributionally Robust Optimization (DRO) strategy. The MoE consists of multiple expert networks, each specializing in different MILP distributions. For each problem instance, RoME uses a graph neural network (GNN) to extract embeddings and dynamically routes the instance to the most suitable expert. The DRO strategy include inter-domain robustness, achieved by minimizing the worst-case loss across domains, and intra-domain robustness, which stabilizes expert selection through perturbations on task embeddings. This approach allows RoME to adapt to the unique structural characteristics of each instance and improve generalization across domains.

- Experiments shows that RoME significantly improves generalization and performance across multiple domains, including unseen ones, achieving a 67.7% average improvement over strong baselines.

**Questions:**

Please see "Strengths And Weaknesses"

**Ethical Concerns:**

["NO or VERY MINOR ethics concerns only"]

**Final Justification:**

Authors addressed my concerns during rebuttal.

**Limitations:**

Please see "Strengths And Weaknesses"

**Quality:**

3

**Strengths And Weaknesses:**

Overall, I think this is a good paper. It demonstrates the efficacy of ML-model-based methods to significantly boost the runtime of  traditional solvers.

Strengths:
- Experimental results are convincing. RoME demonstrates strong generalization capabilities across multiple domains, including unseen ones, which is a significant improvement over existing methods (e.g. ConPS).

- The use of a two-level DRO strategy is quite neat and, according to the ablations in D.4, it looks like it indeed enhances the model's robustness to distributional shifts, improving its adaptability to new problem instances.

- The MoE architecture allows for specialization among experts, leading to better performance on diverse MILP distributions.

Weaknesses and Limitations:

- The complexity of the model may pose challenges in terms of interpretability and understanding the decision-making process of the experts. Additionally, the model's performance heavily relies on the quality of the task embeddings and the effectiveness of the routing mechanism.

- The need for a large and diverse training dataset to fully leverage the cross-domain capabilities of RoME could be a barrier in some applications.

- The approach may not be easily applicable to domains with limited data availability or where the problem structure is not well-represented in the training set.

- The method's effectiveness on domains with drastically different structures from the training set remains to be fully explored.

---

> ### Author Rebuttal · Authors · 2025-07-31
>
> Dear Reviewer SAaz,
>
> Thank you for your kind support and valuable comments. We sincerely hope our rebuttal adequately address your concerns. If so, we would deeply appreciate it if you could consider raising your score. If not, please let us know your further concerns, and we will continue actively responding to your comments.
>
> **W1 Model interpretability & performance**
>
> > The complexity of the model pose challenges for interpretability.
>
> We appreciate this concern. Model complexity is indeed a common challenge for most learning algorithms. Below we address your concerns in detail.
>
> **1.RoME is designed as a pretrained model for MILP solution prediction, not merely a task-specific predictor.**
>
> Unlike prior works such as PS [1] and ConPS [2], which are tailored to a single domain, **RoME is a pretrained model and targets multiple domains simultaneously**. Although this slightly increases model complexity, it also delivers **substantial performance gains across domains**. We analyze the reasons for the success of RoME below.
>
> - **Scalability is essential for a cross-domain MILP model.**  Inspired by progress in large language models, computer vision, and drug discovery, we adopt a MoE architecture. The **GNN encoder** is identical to earlier work [1-4] and remains the primary computational cost. We add multiple lightweight MLP-based experts and decoder heads, so these introduce minimal inference overhead. Here we provide an intuitive comparison between prior work and our approach. **RoME has 697.89K parameters (storage 2.66MB), compared to 449.05K parameters (storage 1.71MB) in PS** [1], showing only a slight increase in model size. Meanwhile, our approach achieves substantial gains in cross-domain generalization ability. In addition, as **Fig. 1(b) in the main text** shows, scaling the number of experts together with the number of training tasks **consistently improves performance**.
> - **RoME meets the three pillars of a foundation model.** In recent years, foundation models have emerged across a wide range of fields. Empirically, their success relies on three core components: a robust training approach, a scalable architecture, and access to large-scale data. (i) *Training approach*: we use a **two-level DRO objective** to balance inter- and intra-domain robustness. (ii) *Scalable architecture*: the **MoE** design enables the model to **perceive and differentiate** diverse domains. (iii) *Large training data*: RoME holds the ability to scale along with the large data. Our work represents the first step toward a foundation model in the MILP domain. While still in its early stages, we hope it lays the groundwork for future development in this direction.
> - **Lower overall cost compared with single-domain baselines.** Once **pre-trained**, RoME can be reused for any target domain without retraining, whereas previous methods must be **re-trained from scratch** for each new domain and degrade sharply outside it. In practice, this makes RoME ***more economical*** despite its larger architecture.
>
> **2.RoME learns a deeper understanding in the routing mechanism.**
>
> Current works typically adopt GNN-based models, which are inherently treated as black-boxes. However, with the recent progress in large language models, an increasing number of interpretability techniques have emerged, particularly for MoE architectures, offering new opportunities to better understand model behavior. Following this, we have presented an interpretability study in **Section 4.4** of the main text. Below we give a further analysis.
>
> - **Expert-activation patterns across tasks.** We examine which experts are activated for different tasks and find that **tasks with similar structure consistently select the same subset of experts**. This indicates that each expert has learned **intrinsic cross-domain patterns** rather than memorizing single tasks.
> - **Latent-space visualization of task embeddings.** We project the learned task embeddings into 2-D space (T-SNE). **Structurally similar tasks cluster tightly together**, confirming that RoME captures correct task representations. This, in turn, shows that the routing mechanism maps tasks to experts in a coherent, task-aware manner.
>
> > The model performance heavily depends on routing & task embedding quality.
>
> We acknowledge that **routing precision and task embedding quality are the core targets of our intra-domain DRO strategy, which acts as the bridge between training and architecture.** By perturbing task embeddings during training, intra-DRO properly activates the routing gate, ensuring that only the appropriate experts are activated—even under small representation shifts. This is pivotal for RoME’s cross-domain generalization; its effectiveness stems from the joint design of our DRO training strategy and MoE architecture, as detailed in **Appendix D.4**.
>
> **W2 & W3.1 Data efficiency**
>
> > RoME requires more data for training.
>
> In fact, in data-limited scenarios, our approach demonstrates stronger performance. This is because **RoME is a pretrained model** that can be trained on diverse synthetic datasets to learn transferable, cross-domain patterns. In contrast, prior methods are trained on task-specific data and can only be applied to the corresponding target domain. Unfortunately, collecting sufficient training data for each new domain is often impractical. Additionally, to further demonstrate the **scalability** of our approach, and in line with your suggestion, we have conducted the following analysis. We trained **RoME** and the **PS** baseline using 30 %, 50 %, and 80 % of the original training data, then evaluated both models on the widely adopted **Set Cover** benchmark. The results (see the Table 1) show that RoME consistently surpasses PS at every data budget, and the performance gap **widens as the data volume increases**.
>
> Table 1: Performance of each method with the limited data.
>
> |             | 30%        | 50%        | 80%        |
> | ----------- | ---------- | ---------- | ---------- |
> | PS+Gurobi   | 125.29     | 125.22     | 125.20     |
> | RoME+Gurobi | **125.21** | **125.14** | **125.11** |
>
> In **Section 4.3 and Appendix D.1**, RoME takes a large step forward by delivering **zero-shot** performance on the challenging industrial benchmark **MIPLIB**, whereas previous methods first have to build and train on a dedicated MIPLIB dataset—often infeasible in practice. To mirror real-world constraints more closely, we follow [3]: (i) we test RoME in a data-scarce setting where only **80%** of the original IIS training data are available, and (ii) we fine-tune RoME on the training data constructed in [3]  while freezing the GNN encoder and finetuning only the expert and head networks. With **fewer than 10 steps**, RoME already surpasses training-specific baseline such as PS. The complete results are reported in Table 2.
>
> Table 2: Performance of finetuning on the IIS datasets.
>
> |           | BKS  | PS+Gurobi | RoME zero-shot+Gurobi | RoME finetune+Gurobi |
> | --------- | ---- | --------- | --------------------- | -------------------- |
> | ex1010-pi | 233  | 241       | 241                   | **239**              |
> | fast0507  | 174  | 179       | 179                   | **174**              |
> | ramos3    | 186  | 225       | 233                   | **222**              |
> | scpj4scip | 128  | 133       | 133                   | **132**              |
> | scpl4     | 259  | 275       | 279                   | **274**              |
>
> **W3.2 & W4 More complicated problem structure**
>
> > RoME should explore more complicated problem structure during training.
>
> We sincerely thank you for your insightful suggestions! As illustrated in **Figure 1(a)** of the main text, the key difference between prior works and **RoME** lies in the **training domains**. Compared to previous methods [1–4], our approach takes a significant step forward by being trained on **diverse domains**, each with distinct problem structures, whereas prior works are trained on a **single fixed domain**. For example, in our main experiments, RoME is trained on three different MILP families, each representing a unique problem structure. During training, we randomly sample instances from these families, meaning that the **problem structure dynamically varies across training batches**. In contrast, prior methods are trained on data from a single domain, where the structural pattern remains constant. Moreover, to evaluate **cross-domain generalization**, we not only test RoME on **unseen synthetic domains**, but also evaluate it on the **real-world MIPLIB benchmark**, which includes a wide variety of industrial instances spanning domains such as **manufacturing, energy, and transportation**. To give a clearer picture of the structural diversity, we list the **constraint types** of a subset of MIPLIB instances used in our experiments, as shown in **Table 3**.
>
> Table 3: Constraints type of different benchmarks.
>
> | Benchmark          | Constraints type                                             |
> | ------------------ | ------------------------------------------------------------ |
> | IP                 | knapsack                                                     |
> | SC                 | set_covering                                                 |
> | bab2               | aggregations, set_partitioning, set_packing, set_covering, cardinality, invariant_knapsack, equation_knapsack, knapsack, mixed_binary |
> | neos-3754480-nidda | precedence, variable_bound, knapsack, mixed_binary           |
>
> [1] A gnn-guided predict-and-search framework for mixed-integer linear programming. ICLR 2023.
>
> [2] Contrastive Predict-and-Search for Mixed Integer Linear Programs. ICML 2024.
>
> [3] Apollo-MILP: An Alternating Prediction-Correction Neural Solving Framework for Mixed-Integer Linear Programming. ICLR 2025.
>
> [4] Differentiable Integer Linear Programming. ICLR 2025.

---

### Note · Authors · 2025-08-16

Dear Area Chair and Reviewers,

Thank you for the thorough and constructive review process. Your feedback has been invaluable in strengthening our paper.

Based on the reviewers' feedback, the paper's key strengths are as follows.

- Reviewers consistently recognized the novelty and motivation of RoME’s design as a significant contribution.

   ○ RoME was described as **“a good paper”** with **“convincing results”** and a **“neat two-level DRO strategy.”** (`Reviewer SAaz`)

   ○ The MoE structure was noted to **“allow for specialization among experts, leading to better performance on diverse MILP distributions.”** (`Reviewer SAaz`)

- The method demonstrated strong improvements in cross-domain performance.

   ○ RoME achieves a **“67.7% average improvement over strong baselines”** on unseen domains. (`All reviewers`)

   ○ Reviewers highlighted RoME’s **“strong generalization capabilities”** and **“significant improvement over existing methods.”** (`All reviewers`)
- The clarity of the paper was also commended.

   ○ The paper was described as **“well-written, technically sound, and extensive in experiments.”** (`Reviewer 4Hsn`)

   ○ It was noted that **“all proposed components contribute to performance.”** (`Reviewer SKRg`)

In response, the authors provided a comprehensive rebuttal that went beyond clarification by conducing extensive new experiments and giving detailed explanation to address every major point.

- Providing more insights into RoME’s behavior and explaining why the architecture generalizes well across domains. (`for all reviewers`)

- Adding detailed quantitative evaluations, including win-count and mean-gap metrics, to better compare performance against Gurobi. (`for Reviewer 4Hsn`)
- Conducting broader experiments on harder benchmarks with integer and continuous variables, as well as extensive ablation studies on $k_0, k_1, \Delta$, and additional runs on all five datasets in the main paper to address reviewer's further concern. (`for Reviewer i5gC`)
- Justifying the contribution of the fuzzy-consistency loss through ablations and theory, and enhancing clarity with expert activation visualizations. (`for Reviewer SKRg and SAaz`)

We are fully committed to incorporating all new results and clarifications into a significantly improved final manuscript. We will also fully **open-source our code** to ensure complete reproducibility.

Thank you again for your consideration.

Best,

Authors

---

### Decision · Program_Chairs · 2025-09-17

**Decision:**

Accept (poster)

**Comment:**

(Partial) solution prediction is one of the paradigms in speeding up MILP solving. However, prior methods typically focused on learning on particular domains of MILPs, limiting the general applicability of the learnt models. In this paper, the authors propose to use a mixture-of-experts architecture to learn multiple domains simultaneously, with a routing mechanism based on learnt task embeddings. Beyond being able to handle the learnt domains, the resulting model also demonstrates generalization to other domains not in the training set.

Reviewers and I generally find the result relevant to the NeurIPS community, and the experiments show promise against prior methods. The rebuttal/discussion also resolved reviewer concerns. I am happy to recommend acceptance.